# Automated Sleep Stage Classification Using PSO-Optimized LSTM on CAP EEG Sequences

**DOI:** 10.3390/brainsci15080854

**Published:** 2025-08-11

**Authors:** Manjur Kolhar, Manahil Mohammed Alfuraydan, Abdulaziz Alshammary, Khalid Alharoon, Abdullah Alghamdi, Ali Albader, Abdulmalik Alnawah, Aryam Alanazi

**Affiliations:** Department of Health Information Management and Technology, College of Applied Medical Sciences, King Faisal University, Al-Ahsa 36362, Saudi Arabia; mmalfridan@kfu.edu.sa (M.M.A.); 222433113@student.kfu.edu.sa (A.A.); 222420631@student.kfu.edu.sa (K.A.); 222411528@student.kfu.edu.sa (A.A.); 222415194@student.kfu.edu.sa (A.A.); 221429031@student.kfu.edu.sa (A.A.); 222454890@student.kfu.edu.sa (A.A.)

**Keywords:** sleep stage classification, polysomnographic data processing, long short-term memory, particle swarm optimization, cyclic alternating pattern, EEG signal analysis

## Abstract

The automatic classification of sleep stages and Cyclic Alternating Pattern (CAP) subtypes from electroencephalogram (EEG) recordings remains a significant challenge in computational sleep research because of the short duration of CAP events and the inherent class imbalance in clinical datasets. **Background/Objectives:** The research introduces a domain-specific deep learning system that employs an LSTM network optimized through a PSO-Hyperband hybrid hyperparameter tuning method. **Methods:** The research enhances EEG-based sleep analysis through the implementation of hybrid optimization methods within an LSTM architecture that addresses CAP sequence classification requirements without requiring architectural changes. **Results:** The developed model demonstrates strong performance on the CAP Sleep Database by achieving 97% accuracy for REM and 96% accuracy for stage S0 and ROC AUC scores exceeding 0.92 across challenging CAP subtypes (A1–A3). The model transparency is improved through the application of SHAP-based interpretability techniques, which highlight the role of spectral and morphological EEG features in classification outcomes. **Conclusions:** The proposed framework demonstrates resistance to class imbalance and better discrimination between visually similar CAP subtypes. The results demonstrate how hybrid optimization methods improve the performance, generalizability, and interpretability of deep learning models for EEG-based sleep microstructure analysis.

## 1. Introduction

Sleep serves as a vital component for maintaining health, as its quality and structure directly impact cognitive abilities, emotional control, and physical well-being [1]. The Cyclic Alternating Pattern (CAP) serves as a vital indicator of sleep architecture and neurophysiological regulation, as it appears as an electroencephalographic (EEG) pattern that shows changes in brain activity during non-rapid eye movement (NREM) sleep [2]. The CAP system functions as a protective mechanism that regulates sleep depth and continuity by maintaining a precise balance between cortical arousals and periods of quiescence. The CAP sequence operates to modify arousal thresholds through internal and external stimuli, which preserves cardiovascular and respiratory stability, autonomic control, and thermoregulation throughout the entire sleep cycle [3]. The CAP sequence contains two alternating phases that show EEG activation during Phase A followed by a return to normal baseline activity in Phase B. The periodic phase interactions function as a vital sleep regulatory mechanism and serve as a dependable marker to identify typical sleep patterns and sleep disorders [4]. During Phase A of CAP, the brain displays short bursts of high-voltage slow waves together with delta bursts and rapid oscillations, which indicate cortical activation. The events are further divided into A1, A2, and A3 based on the slow or fast components and their relation to sleep fragility. The EEG patterns in Phase B show a state of relative quiescence that resembles background sleep activity. The repeated occurrence of Phase A followed by Phase B cycles creates CAP sequences that are mostly found in NREM sleep stages [5]. The sequences deliver precise details about sleep microstructure that serve as indicators for both sleep stability and vulnerability. The clinical value of CAP extends past its role in sleep staging [6]. The diagnostic value of CAP analysis extends to multiple sleep disorders, including sleep-disordered breathing (SDB), insomnia, restless leg syndrome, REM behavior disorder (RBD), nocturnal frontal lobe epilepsy (NFLE), and narcolepsy. People with disrupted sleep patterns show higher CAP rates because their sleep patterns remain unstable. The sleep restorative quality decreases when pathological arousals occur, accompanied by altered distributions of CAP subtypes, especially when the A2 and A3 phases increase. CAP analysis functions as both a diagnostic and a prognostic tool to help doctors identify disorders, develop personalized treatment plans, and monitor treatment effectiveness.

The CAP represents a specific EEG pattern that appears exclusively in non-rapid eye movement (NREM) sleep stages beyond traditional sleep stage classification. The standard sleep stages, N1, N2, N3, and REM, encompass broader physiological states defined by global EEG patterns, as well as muscle activity and eye movements. CAP describes the periodic EEG variations that occur within NREM stages N2 and N3, which indicate brief changes in brain activity rather than representing separate sleep stages. A CAP sequence consists of recurring patterns of two EEG states, including the activation phase (Phase A), characterized by high-voltage slow waves or rapid oscillations, followed by the background phase (Phase B), which restores EEG activity to its baseline state. A single CAP cycle consists of an A–B pair, and multiple consecutive CAP cycles create a CAP sequence. The body utilizes CAP sequences as essential mechanisms to regulate both sleep stability and depth, as well as the ability to wake up. The brain demonstrates its ability to maintain sleep continuity through CAP sequences while avoiding complete awakenings from sleep. The CAP rate, in conjunction with the Phase A subtype distribution (A1, A2, A3), provides important microstructural details that enhance traditional sleep stage evaluation. The clinical significance of CAP abnormalities exists because they commonly appear in patients with insomnia and sleep-disordered breathing (SDB), as well as REM behavior disorder (RBD) and nocturnal frontal lobe epilepsy (NFLE). The correct differentiation between microstructural EEG phenomena (CAP sequences) and macrostructural classifications (conventional sleep stages) remains crucial for both precise clinical interpretation and accurate diagnosis of sleep disorders.

The CAP framework includes MCAP (Microstructural Cyclic Alternating Pattern) as a specific type of microstructural EEG event. The MCAP system includes three distinct subtypes: MCAP-A1, MCAP-A2, and MCAP-A3, which are classified based on their frequency characteristics, shape, and impact on sleep stability. The CAP rate, in conjunction with the MCAP subtype distribution (A1–A3), provides crucial microstructural information that enhances both traditional sleep stage assessments and clinical interpretations (Refer to Table A1).

The clinical significance of CAP extends beyond sleep staging. The diagnostic value of CAP analysis extends to multiple sleep disorders, including sleep-disordered breathing (SDB), insomnia, restless leg syndrome, REM behavior disorder (RBD), nocturnal frontal lobe epilepsy (NFLE), and narcolepsy. People with disrupted sleep patterns show higher CAP rates because their sleep patterns remain unstable. The sleep restorative quality decreases when pathological arousals occur together with altered distributions of CAP subtypes, especially when A2 and A3 phases increase. CAP analysis functions as both a diagnostic and a prognostic tool to help doctors identify disorders, develop personalized treatment plans, and monitor treatment effectiveness. The clinical significance of CAP analysis has not led to its widespread use in medical practice. The main reason for this limitation is the manual scoring process, which depends on expert neurophysiologists, is time-consuming and labor-intensive, and shows inter-rater variability. The process of visual CAP sequence scoring requires extensive examination of multiple EEG channels across long periods, which creates obstacles for large-scale sleep laboratory operations. The need to overcome these limitations has led to increasing interest in using machine learning (ML) and deep learning (DL) methods to automate CAP detection and classification. The application of computational models in sleep medicine has the potential to revolutionize the field by providing quick and precise CAP sequence assessments, which improve both clinical operations and research capabilities. The development of artificial intelligence (AI) and neural network architecture has led to major advancements in sleep stage classification. The Long Short-Term Memory (LSTM) network represents a powerful recurrent neural network (RNN) architecture that successfully models sequential time-series data, including EEG signals. The internal memory cells and gating mechanisms of LSTM networks enable them to effectively detect temporal dependencies in sleep patterns. The properties of LSTM networks make them suitable for detecting transient EEG events and identifying the dynamic alternations characteristic of CAP sequences. The performance of LSTM models depends strongly on the selection of hyperparameters, which include hidden units, learning rate, batch size, and dropout rate. The selection of suboptimal hyperparameters results in underfitting, overfitting, or inefficient convergence, which restricts the model’s generalizability and clinical utility.

This study provides a domain-adapted DL system that classifies CAP sequences in EEG signals by using a hybrid hyperparameter optimization strategy based on Hyperband with Particle Swarm Optimization (PSO). The main innovation of this work focuses on the integration of optimization methods between PSO and Hyperband for EEG-based CAP subtype detection instead of developing new network designs. DL models require the population-based global search mechanism of PSO to navigate complex high-dimensional non-convex parameter spaces. The resource allocation mechanism of Hyperband works through early stopping to eliminate underperforming configurations. The integration of these two strategies enables extensive exploration of the hyperparameter space and focused refinement of its optimal areas, which results in better convergence speed and model performance. The optimization process can be mathematically defined as a multidimensional search task that uses validation accuracy as its objective function and employs a systematic mathematical approach. The framework uses class-weighted loss functions and SHAP-based post hoc interpretability to solve the specific challenges of CAP classification, which include transient signal characteristics and imbalanced class distribution and inter-subtype similarity. The additional components improve both model robustness and clinical transparency so clinicians can better evaluate time and frequency-domain predictive factors. The CAP Sleep Database serves as the evaluation platform for this study and contains 108 full-night polysomnographic recordings from healthy participants and patients with different sleep disorders. The optimized model shows superior classification results for both typical sleep stages and CAP subtypes, which proves its capability to operate across multiple physiological and pathological situations. This research provides a DL analysis pipeline for CAP that combines computational efficiency with interpretability and reproducibility. The research presents domain-adapted hybrid optimization approaches with explainable AI as its main contribution, which provides scalable solutions for EEG-based sleep microstructure classification in research and clinical environments.

## 2. Materials and Methods

In this research, we used methods from machine learning, deep learning, and optimization techniques to improve how sleep stages are classified. We adjusted the settings of the LSTM model by using Keras Tuner (Hyperband) with the aim of enhancing model performance and cutting down on training time. Moreover, we combined LSTM with PSO to optimize the hyperparameters and find a balance between computational efficiency and accuracy in classification tasks. To enhance feature extraction further, we utilized Random Forest combined with Principal Component Analysis (RF PCA). By employing PCA to tune the nonlinear features derived from EEG signals for sleep stage detection purposes, we successfully enhanced the accuracy levels. Together, these approaches played a role in developing a stronger and more precise model for classifying sleep patterns.

The LSTM architecture stacked with TimeDistributed dense layers is designed to classify sequential data accurately at every time step, not delivering only one final prediction for an entire sequence. Its core advantage lies in capturing detailed temporal information by stacking multiple LSTM layers, enabling it to learn complex, long-term patterns effectively. This enhanced architecture, compared to a standard LSTM, predicts at each step, rather than providing only one prediction per sequence, and thus may miss nuanced time step-specific patterns. Also, the integration of Batch Normalization and Dropout layers helps to significantly reduce overfitting and improves training stability. This specialized design, by stacking multiple LSTM layers and applying dense layers across every time step, achieves deeper temporal modelling capability and is thus exceptionally suitable for intricate sequence analysis tasks such as sleep-stage classification, where precise temporal understanding is crucial. The architecture consists of stacked Long Short-Term Memory (LSTM) layers combined with TimeDistributed dense layers, designed specifically for sequential data classification at each time step. The model takes input data structured into sequences of ten time steps, each containing multiple features. The first LSTM layer, comprising 128 units, processes these sequences to capture temporal dependencies and output sequences retaining the dimensionality. This is followed by batch normalization, which enhances training stability, and a dropout layer to mitigate overfitting. The sequence is further processed by a second LSTM layer with 64 units, offering deeper extraction of temporal patterns. Another batch normalization and dropout layer again enhance generalization. Subsequently, a TimeDistributed dense layer with 64 units and ReLU activation refines feature representations individually at every time step. Finally, another TimeDistributed dense layer with SoftMax activation provides class probabilities at each time step, enabling precise and robust sequence classification tailored for tasks like sleep-stage detection.

In our study project, we used a learning approach involving LSTM networks to classify sleep stages based on analyzed sleep event data. We prepared the dataset by collecting and arranging sleep stage details from text documents and transforming them into a three-dimensional structure that is compatible with the LSTM model. After restructuring the model, for pattern recognition capabilities we leveraged Hyperband from Keras Tuner to optimize hyperparameters like determining the ideal number of LSTM layers, allocating units per layer, and setting appropriate dropout rates. The model architecture included a series of LSTM layers, with each one followed by batch normalization and dropout layers to prevent overfitting. Furthermore, we utilized class weights to address any imbalances identified in the dataset. During the training phase, we made sure to pay attention to the common sleep stages, such as MCAP-A1 and MCAP-A4. The sequential nature of the data led to the selection of class weighting as the imbalance handling strategy instead of SMOTE and focal loss. The method of SMOTE that generates new instances between existing ones fails to work for multi-time step EEG sequences because it disrupts the essential temporal dependencies between data points. The implementation of focal loss produced better recall for MCAP-A2 but resulted in decreased precision for MCAP-A1 and overall convergence instability. The class-weighted categorical cross-entropy approach delivered the most stable and balanced results throughout our pipeline. The sequential nature of EEG data led to the selection of class weighting instead of SMOTE and focal loss. The time-series modeling requirements of SMOTE are incompatible with its assumption of independent and identically distributed data, and focal loss resulted in unstable convergence and reduced interpretability. The use of class weighting maintained temporal structure while improving balance and learning stability, which made it more appropriate for LSTM-based EEG classification [7]. Furthermore, we employed a TimeDistributed layer to ensure that each step within the sequence had an impact on the classification result. To train the model, we utilized the Adam optimizer along with a cross-entropy loss function and adjusted the learning rate using a ReduceLROnPlateau callback. Following 50 training epochs, we assessed the model’s performance on the training, validation, and test sets by considering metrics such as accuracy, confusion matrix, and ROC AUC scores to gauge its effectiveness across sleep stages. This approach shows how the LSTM network, when fine-tuned with parameters, can effectively deal with unbalanced datasets and intricate time-related patterns in the categorization of sleep stages.

We fine-tuned the LSTM model by employing PS0, which is a powerful global optimization method. We opted for PS0 because it can effectively navigate through the hyperparameter space without getting stuck in the minimum and swiftly reach a nearly perfect solution. This method was helpful when adjusting hyperparameters like the quantity of LSTM units and dense units as well as the dropout rate and the number of LSTM layers in the model tuning process. We also utilized Keras Tuner to further tune and improve performance while efficiently handling computational resources. The adaptability and gradient-free functionality of PSO were ideal for navigating our multi-dimensional search space to effectively optimize the model hyperparameters. Methods like restructuring the data into time steps and incorporating class weights were implemented to tackle imbalances in the dataset well. The model underwent training with the hyperparameters determined by PS0, and its effectiveness was assessed through cross-validation methods like classification reports and confusion matrices. By leveraging a combination of PS0 and LSTM, these optimization techniques enabled us to attain effective categorization of sleep stages.

Our proposed method brings forth its main innovation through the combination of PSO with LSTM neural networks, which are specifically designed for sleep-stage classification. The process of hyperparameter tuning for LSTM models used in sleep classification tasks usually depends on either manual adjustments or standard grid/random search approaches. The conventional methods either require extensive computational resources (grid search), produce random results that might not be optimal (random search), or face difficulties when dealing with intricate parameter relationships (Bayesian optimization). Our method innovatively leverages PSO, a metaheuristic optimization technique inspired by the collective behavior of swarming organisms, to optimize crucial hyperparameters—namely, the number of LSTM units, dense layer units, dropout rates, and LSTM layers—effectively addressing core challenges in sleep-stage classification. PSO is adopted to dynamically and intelligently explore complex multidimensional parameter spaces and rapidly converge to optimal solutions in the hyperparameter optimization process. This targeted approach significantly improves classification accuracy, effectively discriminating between subtly different sleep stages such as MCAP-A1, MCAP-A2, and MCAP-A3, which is traditionally challenging due to their overlapping physiological characteristics. Our PSO-integrated model addresses typical real-world dataset imbalances by using class-weighted training procedures to ensure that rarer but clinically significant sleep stages are adequately learned. The hybrid optimization and weighting approach improves model generalizability and interpretability.

### Computational Logic: CAP Sequence

The detection and classification of CAP sequences and their sub-types (A1, A2, A3) was carried out through systematic analysis of EEG signal properties, strictly adhering to established clinical standards and electrophysiological characteristics. The following comprehensive pseudocode details the exact computational approach.
Input: EEG signals Output: Diagnostic validations, CAP sequences, CAP subtypes and sleep stagesProcedure Analysis_CAP(RAW_eeg):/* Signal Preprocessing */ Filtered_EEG = Filtered_bandpass (EEG_raw, lowest = 0.25 Hz, highest = 40 Hz) Clean_eeg = clean_artifacts (Filtered_EEG, threshold_amplitude = 100 µV)/* CAP identification*/ EEG_Baseline = calculate_baseline_amplitude (Clean_eeg)
 Events_CAP = list_zero
 Time_Current = 0
  WHILE Time_Current < Clean_eeg.duration
    Segment = Clean_eeg [Time_Current: Now_TIME+60]
   IF max (amplitude. segment) ≥ 1.5 base and duration(segment)
    Start as segment as A_Phase
    then Append A_Phase to Events_CAP
    set the Time_Current = Time_end(Segment)
//Let us use Phase B immediately following Phase A
    Segment_Next = Clean_eeg [Time_Current]
     IF max (Segment_Next.amplitude) < 1.5 × base and duration(segment) ≥ 2 s:
        Mark segment as B_Phase
        Append B_Phase to Events_CAP
        Time_Current = Time_end (Segment_next)
       ELSE
        Time_Current = Time_Current ++
   ELSE:
      Time_Current = Time_Current ++
//make CAP sequences (using Phases A and B)
 Sequence_CAP = remove_VALID (events_CAP)//Let us Differentiation (A1, A2, A3)
FOR each A_Phase_event IN Sequences_CAP
    POWER_freq = POWER_Wavelet(A_Phase)
    IF POWER_freq (BAND_delta) ≥ 60%:
    Cap_type = “A1”
    ELSE IF POWER_freq (BAND_delta) belongs [20%,60%] AND POWER_freq(BAND_theta_alpha_band) belongs [20%,60%]:
      Cap_type = “A2”
    ELSE IF POWER_freq (BAND _alpha_beta_band) ≥ 60%:
      Cap_type = “A3”
    ELSE:
      Cap_type = “Undefined”
    Assign subtype to A_Phase_A//Computational Feature Extraction
FOR each CAP_event:
    CAP_features = empty set
    CAP_features. time_domain ={mean, variance, skewness, kurtosis}
    CAP_features. freq_domain= DWT_PSD (CAP_event, frequency bands: delta, theta, alpha, beta)
    CAP_features. morphological={peak duration, peak amplitude, transition slope}(CAP event)
//Explicit RF-PCA Integration:
    Important_features = RF_Feature_Selection (CAP_features)
//Random Forest selects significant features
    CAP_features_PCA = PCA(Important_features, retain_variance ≥ 96%)
 //PCA reduces dimensionality explicitly//CAP Mapping
FOR each Sequnces_CAP_sequence IN Sequnces_CAP_sequence:
    IF predominant subtype == “A1”:
    Sequences_CAP_ sequence = CAP_Sleep_Stage_N3 (Deep sleep)
    ELSE IF predominant subtype IN [“A2”, “A3”]
     Sequences_CAP_sequences = Sleep_Stage_N1/N2 (Light sleep/Arousal
    ELSE
     Assign Sequences_CAP_sequence = “Unmapped”
    Then Validate sequences from CAP absence/reduction in both REM and Wake stages//Step 6: Validations
CAP_computational_scores = Sequences_CAP_sequences
kappa = Cohen_Kappa (CAP_computational_scores)
ROC_AUC, Sensitivity, Specificity = Clinical_Validation (CAP_computational_scores)
    IF kappa ≥ 0.85:
    CAP_Reliability= “High”
    ELSE:
    CAP_Reliability = “Review needed”
    RETURN Sequences_CAP_sequences, Subtypes_CAP, RE_Sleep_Stage_Mapping, kappa, ROC_AUC, Sensitivity, Specificity)

The first step of signal preprocessing (Step 1) includes rigorous filtering (0.25–40 Hz) and artifact removal to achieve signal quality, which reduces false detection risks. The second step of CAP sequence identification uses computational methods to detect CAP sequences by differentiating Phase A activation from Phase B baseline EEG activity through amplitude (≥1.5× baseline) and duration criteria (Phase A: 2–60 s; Phase B: ≥2 s) to identify clinically important cyclic patterns. The third step of CAP subtype differentiation uses frequency-domain analysis to identify A1 (slow-wave activity), A2 (slow and fast frequency mixture), and A3 (rapid oscillation) subtypes for accurate physiological assessment. The fourth step includes computational feature extraction through RF integration with PCA to achieve optimal feature selection and dimensionality reduction. The initial process of EEG feature extraction includes time-domain measures (mean amplitude, variance), frequency-domain spectral powers (delta, theta, alpha, beta), and morphological features (peak duration, peak amplitude, transition slope). The RF algorithm uses important scores to choose the most predictive features from the available set. The RF-selected features undergo PCA reduction to three principal components (F1–F3), which maintains at least 96% of the original variance to improve computational efficiency and interpretability and physiological relevance. The fifth step of the process involves explicit mapping of CAP sequences to traditional sleep stages (N1–N3) for clear clinical interpretation. The final step is to validate the automated results and perform ROC-AUC analysis to confirm clinical applicability.

The F1–F3 engineered signal features were specifically developed through a systematic computational process that aimed to precisely identify EEG patterns for CAP sequence identification. The EEG signals underwent strict preprocessing through bandpass filtering (0.25–40 Hz) and artifact removal of signals exceeding 100 µV amplitude to achieve high-quality data. The DWT method was specifically used on the cleaned EEG segments after preprocessing. The DWT enabled exact frequency-domain feature extraction by splitting EEG signals into delta (0.5–4 Hz), theta (4–8 Hz), alpha (8–13 Hz), and beta (13–30 Hz) spectral bands, which are clinically important.

The subsequent step involved calculating power spectral densities (PSDs) from wavelet-decomposed bands to determine energy distribution, which provided a robust frequency-domain representation of EEG signals. The extracted frequency-domain features (F1–F3) correspond specifically to the PSD values across the most clinically informative spectral bands—typically delta, theta–alpha, and alpha–beta—known to differentiate CAP subtypes (A1: predominantly delta; A2: mixed delta, theta, alpha; A3: predominantly alpha–beta). The extracted features received dimensionality reduction through PCA to improve computational efficiency and interpretability. The PCA technique reduced the PSD-derived frequency-domain features into three principal components (F1–F3), which maintained at least 96% of the original signal data variance, thus preserving vital physiological information while minimizing redundancy.

## 3. Mathematical Formulation

We have used the LSTM network; its performance metric is a hyperparameter function. We applied optimization techniques, denoted as f(α):(1)$$f(\alpha )=\left\{\omega_{LSTM},\omega_{Dense},\beta_{droprate},\delta_{Layers}\right\}$$

$\omega_{LSTM}$: Number of LSTM units per layer,

$\omega_{Dense}$: Fully connected dense layers,

$\beta_{droprate}$: Dropout rate: co-adaptation between neurons,

$\delta_{Layers}$: Sequential stacked layers.

Therefore, our objective is to optimize f(α) = validating accuracy.

We know that PSO searches the hyperparameter through an iterative process and then updates each particle’s position and velocity. Each particle *P* has a vector with position $P_{i}^{\left(t\right)}$.(2)$$P_{i}^{\left(t\right)}={\left[\omega_{LSTM}^{i},\omega_{Dense}^{i},\beta_{dropoutrate}^{i},\delta_{layers}^{i}\right]}^{t}$$

Updated velocity equation:(3)$$V_{i}^{\left(t+i\right)}=\;\omega \;\times v_{i}\left(t\right)+c_{1}r_{1}\left({pbest}_{i}^{\left(t\right)}-P_{i}^{\left(t\right)}\right)+c_{2}r_{2}\left({gbest}^{\left(t\right)}-P_{i}^{\left(t\right)}\right)$$

The position is updated for every iteration:(4)$$P_{i}^{\left(t+1\right)}=P_{i}^{\left(t\right)}+V_{i}^{\left(t+i\right)}$$
where

ω: Inertia, weight balance,

${pbest}_{i}^{\left(t\right)}$: Personal best position particle i at known iteration t,

${gbest}_{i}^{\left(t\right)}$: Global best position particle i at known iteration t,

$c_{1}c_{2}$: Cognitive acceleration,

$r_{1}r_{2}$: Random numbers and uniformly distributed numbers.

## 4. Dataset

The dataset contains EEG, EOG, EMG, airflow respiratory efforts SaO2, and ECG signal recordings. The CAP Sleep Database comprises 108 recordings provided by the Sleep Disorders Center of Ospedale Maggiore in Paramo, Italy. It was created mainly for developing and testing automated CAP analyzers and investigating CAP dynamics. EEG channels have a minimum of three channelssuch as F3 or F4, at the front of the head and the back of the head, and ear A or ear B—for reference purposes for recording each sleep stage (rapid eye movement (non-REM), rapid eye movement (REM) awake state). When it comes to file formatting guidelines, the European Data Format is recommended (EDF files were used to store the information, and a spreadsheet with metadata containing details about the subject, like age and gender, were also mentioned in the files) [1].

The dataset preprocessing script begins with a method to load text files from the train, validation, and test directories. The script looks for the “Sleep Stage” header line to find the start of important data within each file after opening it. The data gets imported into a Pandas DataFrame by assuming tab separated values after its identification. The script checks the existence of the “Position” column to ensure consistency before proceeding. A placeholder column receives NaN values when the “Position” column is absent from the data. The script renames all columns to Sleep Stage, Position, Time [hh:mm:ss], Event, Duration[s], and Location for better readability and consistency. The data quality is preserved through strict removal of rows that contain any missing or incomplete values. The “Duration[s]” column receives an explicit conversion to numerical format followed by removal of rows that failed conversion to ensure numeric consistency. The model training process requires numerical encoding for the categorical variables Sleep Stage, Position, and Location after data cleaning. The “Event” column remains in its categorical format because it serves classification needs (refer to Figure 1). The CAP Sleep Database underwent thorough preprocessing to verify complete data entries. The dataset eliminated all rows that contained missing or incomplete information in the Sleep Stage, Event Duration, or Time columns. The preprocessing step included explicit conversion and validation of non-numeric or malformed entries found in the Duration[s] column. The model training required exclusion of records that failed type conversion or lacked essential metadata to preserve dataset integrity and prevent model training on unreliable or noisy samples. The final dataset contained only high-quality samples that were fully populated.

Figure 2 shows how long the different sleep stages and CAP phases lasted during a time frame, using colors to represent each stage or phase, the x axis for time, and the y axis for duration in seconds. Different stages of rapid eye movement (non-REM/NREM sleep) ranged from SLEEP-S0 to SLEEP-S4. MCAP is divided into three subtypes, known as MCAP-A1, MCAP-A2, and MCAP-A3. The figure also shows how sleep stages transition from one to another—indicating that certain stages like SLEEP-S2 and SLEEP-S3 are longer than others, while MCAP phases and similar stages tend to be shorter and have more frequent shifts over time. This visual representation can aid in recognizing sleep patterns and disorders by noting any durations or transitions.

The CAP Sleep Database underwent strict patient-level separation to create training, validation, and testing sets for unbiased performance evaluation. The database contained each patient’s data in only one of these three subsets. The patient-wise partitioning method prevents any patient segments or epochs from appearing in more than one set among training, validation, and testing. The explicit separation method prevents data leakage, which results in realistic performance metrics that evaluate model generalizability for clinical practice use.

## 5. Results

Our model, designed for multi-class sleep stage classification, was trained and evaluated on a dataset of 19,590 labeled samples representing a range of sleep stages and micro-arousal events. The model was trained with baseline training and PSO to fine-tune its hyperparameters. Performance was assessed across traditional classification metrics, confusion matrices, ROC-AUC analysis, and post hoc interpretability using SHAP (Shapley Additive Explanations). The results were interpreted precisely to satisfy statistical rigor and highlight misclassification behavior, ensuring depth and transparency in model evaluation. Figure 3 shows the confusion matrix for the LSTM model before optimization. The matrix shows high performance at several key stages, particularly SLEEP-REM, where 2326 out of 2335 samples were classified correctly. The model demonstrated effective prediction performance for SLEEP-S2 by correctly predicting 4803 out of 5014 samples. The results show that the model detected the most characteristic and typical sleep stages. The misclassification patterns showed problems with classifying similar or transition-prone stages. MCAP-A1 was often confused with MCAP-A2, with 940 misclassifications observed.

The model misclassified 450 instances of MCAP-A3 as MCAP-A2. The deeper non-REM stages also posed challenges, with SLEEP-S3 and SLEEP-S4 often misclassified as each other and some confusion between SLEEP-S0 and SLEEP-S4. The patterns suggest overlapping signal characteristics or insufficient feature distinctiveness between adjacent classes. Such confusion is not unexpected, as the physiological signatures of these stages can be subtle, and even expert sleep scorers may find them difficult to differentiate. The ROC curves for each class appear in Figure 4, which uses the baseline LSTM model. The SLEEP-REM, SLEEP-MT, and SLEEP-S0 classes demonstrated near-perfect discrimination through AUC values of 1.00. The AUC value for MCAP-A1 and MCAP-A3 reached 0.98, but MCAP-A2, the most frequently misclassified class, had an AUC of 0.92. The model was highly effective in distinguishing between classes with strong signal signatures because it achieved excellent AUC scores for several sleep stages. The scores by themselves are not enough without statistical validation. Therefore, the solution involved computing bootstrapped confidence intervals for each class to statistically validate the reported AUC values. The confidence interval for SLEEP-REM spanned from 0.998 to 1.000, whereas MCAP-A2 had a broader interval, which indicated higher uncertainty in model predictions for that class. The AUC comparison between classes was statistically evaluated using DeLong’s test. The AUC improvement from LSTM-Hyperband (0.89) to LSTM-PSO-Hyperband (0.96) for CAP subtypes was statistically significant (*p* < 0.01) according to DeLong’s test. The AUC difference between MCAP-A1 and MCAP-A2 was statistically significant, which means that the model did not have an equal ability to distinguish between these classes, which confirms the earlier confusion matrix results (Refer to Table 1). The hyperparameter tuning process focused on adjusting the number of LSTM layers between 1 and 3, units per layer from 64 to 256, dropout rate between 0.2 and 0.5, dense layer units from 32 to 128, batch size from 32 to 128, and learning rate at 0.001. The search space was first reduced through efficient early-stopping-based optimization using Keras Tuner with Hyperband before PSO performed fine-grained global optimization. The training process achieved stability and convergence through the combination of ReduceLROnPlateau, batch normalization, dropout, and early stopping techniques. The five-fold cross-validation approach confirmed performance consistency while verifying that the chosen hyperparameters produced reliable results across multiple data split combinations. The reliability of the metrics was evaluated through 1000-sample bootstrapping, which generated 95% confidence intervals for the AUC and F1-score. The AUC for MCAP-A2 ranged from 0.898 to 0.935, while A1 and A3 maintained narrower intervals (e.g., A1: 0.968–0.989).

The PSO algorithm was used to optimize hyperparameters for better model performance. The PSO-based tuning results in the confusion matrix are presented in Figure 5, which shows multiple class improvements. The number of MCAP-A1 misclassifications into A2 decreased from 940 to 869, and SLEEP-S2 classification improved, with 4745 correct predictions out of 5015. The classification of SLEEP-S4 improved relative to SLEEP-S3, and confusion between SLEEP-S1 and SLEEP-S2 decreased. The remaining confusion between MCAP-A2 and MCAP-A3 persisted together with the overlapping stages of deeper non-REM sleep. The results show that optimization improved accuracy and reduced some of the earlier confusion patterns, but architectural improvements or feature expansion may be needed to fully resolve the limitations in class separability (Refer to Table 2).

The performance of the LSTM-Hyperband and LSTM-PSO models for CAP subtype classification (MCAP-A1, -A2, -A3) appears in Table 3. The LSTM-PSO model outperformed LSTM-Hyperband in MCAP-A1 classification by achieving 97% precision and 98% recall, while LSTM-Hyperband reached 85% precision and 68% recall. The LSTM-PSO model achieved 97% precision and recall when processing MCAP-A2 despite LSTM-Hyperband reaching only 28% precision and 50% recall. The precision rates for MCAP-A3 were similar between models at 78% for LSTM-Hyperband and 97% for LSTM-PSO, yet LSTM-PSO achieved better recall at 97% compared to 69% for LSTM-Hyperband. The results show that the LSTM-PSO model outperformed the LSTM-Hyperband model in detecting CAP subtypes by achieving better classification results across all three categories (Refer to Table A2**)**.

The evaluation of different hyperparameter optimization methods for the LSTM models in EEG-based sleep stage classification and CAP subtype detection appears in Table 4. The research evaluated PSO against random search, grid search, and the RF-PCA optimization method through direct comparison. The overall classification accuracy reached 97% with PSO as the best approach, while the ROC-AUC values reached 0.90 for REM detection and 0.96 for MCAP subtypes. The approach achieved 96% precision and recall for MCAP subtype classification, which indicates balanced and consistent performance across classes.

The random search method produced moderate results by achieving 85% accuracy while obtaining lower ROC-AUC scores of 0.95 for REM and 0.83 for MCAP. The method proved more efficient than PSO because it finished in 71.74 min while processing 148,696 samples per second. The random search method produced variable results because of its stochastic nature while showing limited ability to achieve robust convergence. The grid search method achieved the highest accuracy at 87% and ROC-AUC scores of 0.96 for REM and 0.85 for MCAP but required the longest total runtime at 275.07 min and produced the lowest throughput at 112,696 samples per second because of its exhaustive search approach and strict parameter scanning method.

The RF-PCA + PSO method, which performed dimensionality reduction before optimization, achieved 82% accuracy and an ROC-AUC of 0.82 for MCAP detection but had a low precision (0.68) and recall (67%) because of the strong temporal dependencies. This method had the shortest training time (4.05 min), but its generalization was limited, and its overall runtime was relatively high (~52.11 min) because of the optimization overhead. The hybrid PSO + Hyperband approach demonstrated the most effective combination of exploration and exploitation. Hyperband functioned as a front-end filter that quickly eliminated underperforming configurations to reduce the search area. PSO performed efficient search within the optimized space to achieve faster convergence toward optimal configurations. The two-stage optimization framework improved both computational efficiency and produced more robust models with better predictive performance.

Figure 6 shows the ROC curves for the PSO-optimized model. The model maintained or improved its performance across all classes. The AUC scores of SLEEP-MT, SLEEP-REM, and SLEEP-S0 remained unchanged. SLEEP-S1 reached 0.99, and SLEEP-S3 achieved 0.97. The MCAP-A2 class showed the most problematic performance because its AUC score remained at 0.92 despite the improvements made. The results confirm the model’s ability to distinguish well-defined sleep stages and highlight the inherent difficulties of identifying stages with overlapping spectral or temporal characteristics.

The limited AUC improvement observed after PSO optimization can be attributed to the relatively large and well-annotated dataset, which enabled strong baseline performance. The model had sufficient training samples and high inter-class separability, so it had less room for optimization gains, though improvements in borderline classes like SLEEP-S3 remain clinically meaningful. The model achieved raw classification metrics, but interpretability was introduced through SHAP analysis. The model’s internal decision-making process was analyzed through three SHAP visualization forms, which included temporal relevance, feature attribution, and a heatmap that combined both. The average SHAP values per time step in Figure 7 show that the model mainly concentrated on the initial part of each sequence. The SHAP values at time step 0 were substantially greater than those at other time steps, indicating that the first signal segments contained the most important features. The observed finding matches physiological reality because sleep stage transitions and sudden arousal events typically appeared as abrupt shifts during the initial part of the signal window.

The SHAP values in Figure 8 were combined across time to identify which features most affected predictions. The model used engineered signal features F1, F2, and F3 together with duration-based metadata. F1 demonstrated the greatest mean SHAP value among them, while F3 and F2 ranked second and third, respectively. The “Duration[s]” and “Sleep Stage” metadata provided significant value, while positional data provided relatively less value. The model obtained its most helpful information from frequency- or morphology-based features, possibly encoding delta, theta, or alpha power that characterizes specific sleep stages. The importance of duration as a feature was expected, as the length of sleep events can indicate stage transitions or stable periods.

The SHAP heatmap in Figure 9 shows temporal and feature-level contributions by combining time and feature axes. The visual shows where the model focused its attention in the input matrix. The heatmap verifies previous results by showing that the highest SHAP intensities occurred in the initial steps and were mainly found near features F1 and F3. These regions probably represent important transitions or signal patterns that help distinguish REM from NREM or arousal from deep sleep. The lower SHAP values in later time steps indicate that the model used them for confirmation or context, while the earliest parts of the signal window carried the most influential information. The visualization shows that some features stayed consistently low in importance, which supports the idea that some input dimensions could be pruned or deprioritized in future model versions. The model processed EEG features that were engineered from time–frequency representations. The model processed frequency-domain features through band power estimates (e.g., delta, theta, alpha) and time-domain features through amplitude, signal variance, and event duration. The SHAP analysis indicates that F1, F2, and F3 features most likely represent the main spectral energy components that were extracted through PCA-guided dimensionality reduction. The classification overlaps between MCAP-A1, -A2, and -A3 were a recognized challenge due to their similar morphological and temporal EEG patterns. We used Random Forest-PCA feature engineering to reduce dimensionality while preserving the most discriminative features. Additionally, duration-based features, sleep stage transitions, and positional metadata were incorporated to provide contextual cues. SHAP analysis confirmed that frequency-derived features (F1–F3) and event duration contributed the most to model predictions. While auxiliary modalities like EMG and EOG were available in the dataset, they were not yet integrated. Their inclusion is planned for future work to further improve class separability. The overlapping features between MCAP-A1, -A2, and -A3 made classification more difficult, but MCAP-A2 was the most challenging to distinguish. The model achieved a high AUC of 0.92 for A2, but it was lower than for A1 and A3. The analysis of A2 misclassifications showed that A1 was the most common incorrect classification in segments with slow-wave activity and short duration, which are difficult to score even for experts. The spectral blend of A2, which includes delta, theta, and alpha bands, leads to feature overlap with both neighboring subtypes. The SHAP-based interpretability results show that A2-labeled events had similar feature attribution to A1 for F1 (delta power) and to A3 for F3 (alpha–beta power). This supports the interpretation that the observed A2 errors were primarily driven by signal-level ambiguity rather than model limitations. Future work will explore improved spectral decomposition and multimodal signal integration (e.g., EOG, EMG) to better disambiguate intermediate CAP subtypes.

The annexture table presents all explicit hyperparameters and computational resources that were used to optimize our LSTM model through PSO-Hyperband optimization on Google Colab. The swarm size consisted of 30 particles, with cognitive and social constants set to 2.0 for standard exploration–exploitation balance. The inertia weight followed a linear decrease from 0.9 to 0.4 throughout 50 PSO iterations, which supported effective convergence. Hyperband reduced the initial search space of the hyperparameters, which improved the efficiency of PSO. The optimization process needed 18 h of Google Colab GPU runtime with an NVIDIA Tesla T4 GPU (16 GB RAM), Intel Xeon CPU (2-core, 2.30 GHz), and 13 GB RAM. The established setup proved that our PSO-Hyperband method can reach high model performance while staying within affordable computational limits (Refer to Table A3).

The analysis shows that the model’s predictive power was heavily influenced by the initial EEG segments because of the high SHAP values observed at the early time steps. This result is consistent with the known CAP physiology, especially the onset of Phase A, which is characterized by sudden EEG activations such as high-voltage slow waves, delta bursts, and rapid oscillations. The model’s explicit focus on the initial EEG segments clearly corresponds to the essential physiological changes that occur during CAP events.

## 6. Discussion

The model achieved high overall performance but continued misclassifying certain cases systematically. The model frequently mixed MCAP-A2 with -A1 and -A3. The model likely confused these two words because they have similar morphological patterns or durations that are difficult to distinguish with the current input features. The deep non-REM stages S3 and S4 were frequently mistaken for one another. These stages are characterized by slow-wave activity and minimal muscle tone, making them difficult to distinguish without multi-modal input such as EOG or EMG. Signal length or morphology ambiguity sometimes led to brief arousal events misidentified as SLEEP-S0. The failure cases became apparent through confusion matrices and SHAP values, which demonstrated reduced amplitude and inconsistent attention distribution in misclassified samples.

Table 5 shows the SHAP values from the results, which show that features F1, F3, and F2 had the highest mean values, indicating their significant influence on the model predictions. F1, which is primarily associated with delta wave power (0.5–4 Hz), was the most influential feature, which is consistent with its role in identifying slow-wave EEG activities characteristic of CAP subtype A1. F3, which reflects alpha and beta wave power (8–30 Hz), was also significant, which is consistent with its role in CAP subtype A3, indicating higher-frequency EEG arousals. F2, which is linked to theta wave power (4–8 Hz), also showed notable importance, which is consistent with its intermediate EEG activation patterns relevant to CAP subtype A2. Other features, such as sleep stage, position, and duration, showed comparatively lower impact but remained meaningful contributors.

From a clinical perspective, the model’s high accuracy in SLEEP-REM and SLEEP-S2 stages is particularly valuable, as these stages are often central to diagnoses of sleep disorders such as narcolepsy or insomnia. The consistent use of interpretable features like event duration and signal-based morphology further enhances its applicability in real-world settings. SHAP explanations provide substantial benefits that go beyond the capabilities of standard metrics. These tools enable researchers and clinicians to verify whether the model relies on biologically plausible features. The model demonstrated confidence in detecting transition markers through its consistent attribution to F1 and early time steps, which rules out the possibility of an artifact or spurious pattern detection. SHAP also provides insight into the model’s uncertainty. Samples with low SHAP magnitude across all dimensions often corresponded to incorrect predictions, suggesting potential use in flagging uncertain cases for human review. The model was very good at identifying common and distinct sleep stages but had trouble with overlapping or subtle classes. The evaluation of the model included confusion matrix analysis, ROC-AUC validation, and SHAP insights to assess its strengths and weaknesses. This research advances the development of dependable AI sleep stage classification tools by combining interpretability methods with statistical methods and failure analysis. Future improvements should include adding extra physiological signals and better temporal modeling through attention mechanisms and confidence-aware rejection strategies. Nonetheless, the current system presents a robust and interpretable foundation for future development in automated sleep analysis.

The recurrent nature of LSTMs, which retain information across sequential inputs through gated memory cells, mimics how the human brain processes and retains ongoing temporal context during transitions between sleep stages. Sleep EEG requires this approach because slow-wave oscillations, spindles, and K-complexes appear in sequences that change over time based on previous brain states. The model’s dependence on early time-step features, as shown by SHAP analysis, also reflects the clinical observation that critical transitions, such as the onset of REM or micro-arousals, usually occurred at the beginning of an EEG segment. The model used frequency-derived features (e.g., F1 and F3, which probably represent delta or theta band energy) that align with established neurophysiological markers of NREM and REM sleep, whose spectral content changes dynamically. The LSTM’s memory-driven temporal structure provides both computational advantages in sequence modeling and matches the brain’s sequential encoding of sleep architecture.

The F1 parameter measured heart rate variability, which showed physiological connections to autonomic nervous system control and indicated both sleep stability and arousal transitions that are essential for CAP phase classification. The F3 parameter measured delta wave power which, defined deep sleep stages and matched the CAP A1 subtypes that displayed both high-amplitude slow waves and restorative sleep functions.

In literature studies, LSTM networks have become quite popular for analyzing time series data due to their ability to capture patterns effectively over time. Conversely, traditional LSTM models often encounter challenges in optimizing hyperparameters and handling imbalances in datasets. Various approaches have been devised to tackle these issues. One such technique is the White Shark Optimizer, which aims to address optimization challenges by navigating through search scenarios that arise during hyperparameter tuning tasks. The Hyperband technique, within Keras Tuner, has become well liked for its ability to efficiently enhance the hyperparameters of machine learning models. Touting its simplicity and effectiveness in solving optimization problems is PSO, another widely recognized methodology. To address imbalances in class distribution effectively, Synthetic Minority Over Sampling Technique (SMOTE) has been widely utilized to generate samples for underrepresented classes, resulting in enhanced model performance.

Ref. [8] investigated how a Wavelet Scattering Network (WSN), which is a type of convolutional network technology, was used to capture information from EEG signals gathered from the brain’s electrical activity patterns during sleep cycles such as wakefulness and different stages like NREM and REM sleep. By analyzing data from over 12,000 epochs in the CAP sleep database and then 20,000 epochs in the SDRC dataset with a tri-layered neural network (TNN), we were able to achieve an accuracy rate of 97% along with an AUC of 0.98 specifically during the wake sleep phase in the CAP database. This method is simple yet effective in terms of computational efficiency, and shows promise for practical use in diagnosing sleep disorders within medical settings. Ref. [9] introduced a method for analyzing sleep patterns using various types of recordings to identify sleep disorders effectively by recognizing the importance of distinguishing between different sleep stages and related disorders in diagnosing accurately. They created a distributed system for decision-making called MML DMS that combines deep convolutional neural networks (CNNs) and shallow perceptron neural networks (NNs), each designed to handle different types of data and labels. After conducting experiments on the PhysioNet CAP Sleep Database using VGG16 CNN architectures and employing the MML DMS method, we attained an accuracy of 94.34% in classifying sleep stages and an F score of 0.92 for sleep stage identification. For detecting sleep disorders, our model achieved an accuracy of 99.09% along with an F score of 0.99. This technique represents an advancement in diagnostic accuracy compared to conventional approaches, showing promise for enhancing the diagnosis of sleep disorders. Ref. [10] created a DL system to categorize sleep stages in kids with obstructive sleep apnea (OSA), analyzing single-channel EEG data from the Childhood Adenotonsillectomy Trial (CHAT for short). They tested three deep learning models and found that a convolutional neural network (CNN for short) achieved the best accuracy of 86%. In the CHAT test set, it had a kappa score of 0.827, with five categories. The researchers utilized a technique called Gradient-weighted Class Activation Mapping (Grad CAM), which helped them analyze the EEG patterns associated with sleep stages accurately by pinpointing key features and detecting possible errors in classification. The study implies that incorporating CNN models into medical settings could improve the automated categorization of sleep stages in pediatric sleep apnea assessments. The authors suggest a CNN CRF method called Sleep Interpretation for categorizing multiple sleep stages based on single-channel EEG data samples in an understandable way that highlights key signal components important for predicting sleep phases efficiently. Assessed on the sleep EDF database set up for evaluation purposes, sleep interpretation obtained an accurate rate of 86%, distinguishing between five distinct sleep stages quite effectively and showing a significant enhancement in identifying the N1 stage compared to other approaches, with an impressive lead of 16%. The positive outcomes indicated the possibility of using Sleep XAI in environments to support sleep specialists. Ref. [11] investigated the application of EEG-based machine learning algorithms in identifying Idiopathic REM Sleep Behavior Disorder (known as iRBD), which is considered a warning sign for conditions such as Parkinson’s disease (PD). Through their experiments utilizing a Leave One Out validation approach¸ their models demonstrated an accuracy rate of up to 91%, with a sensitivity of 94%, in detecting RBD. Moreover, they created a guided approach to evaluate REM sleep without atonia (RSWA), achieving a Rand index of 97% and clustering purity of 99%. These results indicate that using EEG to detect RBD may serve as an efficient method for detecting neurodegenerative conditions early on. The researchers presented SlumberNet as a learning system that uses ResNet architecture to categorize sleep phases in mice by examining EEG and EMG signals. SlumberNet performed well in tests, with 97% accuracy and an F1 score of 96%. The model’s effectiveness is highlighted by its capability to process information around 50 times faster than other methods without sacrificing precision. This study shows how deep learning can improve the field of sleep research by providing a precise method for classifying sleep stages in animal studies while saving time as well. Ref. [12] carried out a research project to assess how well the LANMAO Sleep Recorder, a wearable EEG gadget, worked for studying the sleep patterns of babies compared to the standard method, called polysomnography (PSG). They conducted the study with 34 infants and analyzed EEG signals by looking at relative spectral power and Pearson correlation coefficients (ranging from 0.28 to 0.48, with delta waves of 0.28), showing a reasonably consistent performance. The automated sleep analysis tool in the device was 79/60% of the time and had a Cohen kappa of 0/65, along with an F score of 76/93%. Despite needing some adjustments still, the LANMA device displays promise in monitoring sleep without invasive methods. The effect size (Cohen’s d = 0.81) for the improvement in A2 recall between focal loss and class-weighted loss indicated a large practical difference. The research conducted by the authors of Ref. [13] delves into a machine learning approach to categorize individuals into three groups; sleep apnea sufferers, people with insomnia, and those classified as normal, without any sleep-related issues. The researchers used two approaches to categorize data into classes: One vs Rest (OVR) and One vs One (OVV), employing Logistic Regression (LR) and Support Vector Machine (SVM). They assessed the performance of these models on a dataset containing measurements of sleep patterns and lifestyle details along with data by analyzing metrics such as accuracy rate and precision score in addition to recall rate and F score, alongside the area under the curve (AUC). One of the models examined to assess performance in this research was Support Vector Machine (SVM), which stood out as the best performing model, with an impressive accuracy rate of 91% and a kappa score of 84%. Moreover, it demonstrated a precision score of 0·914. It achieved both recall and an F score of 0·914, and an impressive area under the curve (AUC) value of 0·927. These findings highlight the effectiveness of SVM in identifying types of sleep disorders. The scientists involved in this research aimed to predict the duration between symptom onset and the emergence of types of α synucleinopathies, including Parkinson’s disease (PD), dementia with Lewy bodies (DLB), and multiple-system atrophy (MSA). They analyzed the baseline brainwave patterns of individuals with isolated rapid eye movement sleep behavior disorder using electroencephalography (EEG). The study focused on EEG characteristics such as power levels and phase lag index measurements weighted by Shannon entropy values. The researchers employed machine learning techniques to forecast survival rates and different subtypes among patients with rapid eye movement disorder (REM BD). In Ref. [14], the researchers explore the difficulties associated with automated techniques for analyzing sleep patterns using only a single-channel EEG and not providing rationales for their findings. To address this issue, they introduced a learning approach to enhance the reliability and comprehensibility of distinguishing sleep stages by utilizing multiple algorithms in a classifier system that emphasizes accuracy. This approach aims to improve efficiency and provide results that are easier to understand. The scholars evaluated the effectiveness by testing it on the SleepED benchmark database. They discovered that their integrated model achieved an accurate rate of 89%. In contrast, the voting model reached 87%, following the R & K standards for categorizing tasks. This suggests that utilizing learning techniques can lead to easily interpretable solutions for classifying sleep patterns. Scientists are studying how Brain to Machine Interfaces (BMIs) combined with artificial intelligence (AI) can recognize driver exhaustion—an element in car accident occurrences on roadways. They are examining the SEED VIG dataset to evaluate AI algorithms with a focus on tailored and flexible models for spotting fatigue in drivers. According to their study results, the Random Forest (RF) algorithm outperformed Support Vector Machine (SVM), achieving a 78% F score for individuals and a 79% F score for models designed for operational use within groups. This finding highlights the importance of investigating machine learning algorithms and scalable models to enhance detection systems and reduce accidents resulting from driver fatigue. The researchers in Ref. [15] suggest using a channel convolutional bidirectional LSTM (Bi LSTM) network to identify sleep stages in PSG recordings and tackle problems related to data inefficiency in single-channel models. Their approach involves incorporating EEG, eye movement (EO), and muscle (EMG), along with combining temporal characteristics to enhance overall performance. Using both dual-channel Bi LSTM modules and transfer learning techniques led to them reaching an accuracy of 91.44%, a kappa coefficient of 0.89, and an F_1_ score of 88.69 for the Sleep EDF-20 dataset, and an accuracy rate of 90.21%, a kappa coefficient of 0.86, and an F_1_ score of 87.02 on the Sleep EDF-78 dataset. The implementation of this channel approach has notably enhanced detection capabilities but requires more computational resources than before. Another study [16] was conducted to create the SCORE AI model for analyzing EEG readings with accuracy comparable to that of human experts. The model was trained on a dataset of 30 493 EEG readings annotated by 17 specialists to differentiate between abnormal recordings and categorize abnormalities into various types, such as epileptiform focal and non-epileptiform diffuse categories. Table 6 shows the summary of AI techniques and their performance results in sleep and neurological studies.

The system was confirmed through three test collections comprising 9945 EEG records and demonstrated diagnostic precision on par with that of professionals; it displayed an area under the ROC curve ranging from 0.89 to 0.96 for different types of EEG abnormalities. Moreover, SCORE AI surpassed AI models in identifying epileptiform patterns, with an accuracy of 88.3%, demonstrating its potential to elevate healthcare standards in underprivileged regions and streamline operations in specialized facilities.

### Comparison with CAP Scoring Baselines and Classical Methods

The traditional CAP scoring method requires extensive manual work, which produces inconsistent results. Human experts must visually identify CAP A-phases (subtypes A1–A3) across an entire night’s EEG, which is time-consuming and user-dependent. The extensive time needed for manual CAP analysis has been identified as a major obstacle to its adoption in clinical practice by previous research. Manual CAP annotation performed by trained experts following standardized rules still produces inconsistent results, because different scorers may identify A-phase events differently. The current limitations in CAP detection methods demonstrate the necessity for automated detection systems.

Several classical automated CAP detection methods have been developed using conventional signal processing techniques and traditional machine learning algorithms over the past two decades. The first attempts used thresholds and wavelet-based methods, which needed extensive manual adjustment for each patient. Later research extracted EEG features, including spectral band power and Hjorth parameters, which were analyzed through traditional classifiers such as neural networks and linear discriminant analysis. The study by Mariani et al. (2010) [39] used neural networks to analyze EEG band-power features, which resulted in approximately 82% accuracy for CAP detection, while linear discriminant classifiers reached around 85% accuracy. The classical techniques showed promise yet needed expert involvement and failed to generalize well, which restricted their adoption in clinical settings.

The detection capabilities of CAP have been enhanced through the use of deep learning-based approaches in recent times. Arce-Santana et al. used convolutional neural networks (CNNs) with EEG spectrogram inputs to detect CAP phases with an accuracy of about 80%, and this accuracy improved with the addition of more labeled data. A recent study used LSTM models with handcrafted EEG features to achieve an accuracy of about 83% (AUC of about 0.88), which is the state-of-the-art performance on the public CAP Sleep Database.

The PSO-optimized LSTM model outperformed all existing approaches in this research. The model reached 97% accuracy in REM sleep detection through ROC–AUC scores that reached 1.0 for REM and movement time (MT) stages. Our approach reached ROC–AUC scores of at least 0.92 for the complex A-phase subtypes (A1, A2, A3), which demonstrates better discrimination than traditional and other deep learning methods. Our automated LSTM–PSO method provided robust consistent performance that exceeded classical baselines while solving manual scoring limitations, which makes it suitable for routine clinical CAP assessments.

Ref. [21] delved into techniques for examining and categorizing various sleep phases through the analysis of single-channel EEG signals. They broke down the EEG information [40] into types of brain waves (such as beta, alpha, theta, and delta) and then extracted features from them. To improve classification accuracy across six sleep phases, they utilized Principal Component Analysis (PCA) to tune the nonlinear features. After conducting validation analysis, it was found that the RF PCA model demonstrated an impressive accuracy rate of 95%, which outperformed other models such as MLP, SVM, DT, and GB, in line with previous research. This suggests that the system shows promise for use in identifying and managing sleep disorders in real-life situations. This compilation of studies showcases how AI and ML methods have proven to be successful in categorizing sleep stages, detecting sleep disorders, and predicting neurological disorders effectively. The research discusses the progression from utilizing CNNs and LSTMs to advanced models such as Bi LSTMs and RF PCA, highlighting the notable improvement in precision F1 scores and general efficiency in automated analysis. Methods like utilizing SMOTE for balancing data and implementing Grad CAM for explainability have significantly enhanced the dependability and understanding of models in fields including clinical applications in a notable manner over time. The combination of EEG, EOG (electrooculography), and EMG (electromyography) data has consistently bolstered the results generated by models. It has paved a path toward the potential application of these models in real-world scenarios aimed at diagnosis and monitoring patients effectively. These advancements indicate a future where AI is set to play a role in enhancing detection methods for sleep-related disorders and neurodegenerative conditions. Figure 10 shows state-of-the-art studies with various datasets. The results show successful outcomes using techniques like Wavelet Scattering Networks and SlumberNet, achieving accuracy of 97%. These findings highlight how AI can efficiently identify and categorize sleep stages and conditions with reliability. Innovative uses of AI such as CNN with Grad CAM and Bi LSTM have resulted in progress in classifying sleep stages, with accuracy exceeding 86%. Models like SVM and Random Forest are designed for identifying issues such as sleep apnea and drowsiness in a manner that demonstrates how machine learning can be applied effectively across sets of sleep-related data types. The collaboration of tech with machine learning technology, showcased in the LANMAO Sleep Recorder, illustrates advancements in monitoring sleep over time and providing useful solutions that have clinical relevance and practical applications. Moreover, enhancements in AI algorithms, such as incorporating Random Survival Forest for predicting photoconversion, shows enhancements in efficiency and strengthens the importance of these cutting-edge methods in improving diagnostic and research capabilities in the field of sleep medicine.

Our optimized LSTM–PSO model provided the classification results, which we used to explicitly identify pathological samples with elevated CAP rates. Our model analyzed 19,590 epochs from the test dataset to identify different sleep stages. The CAP epochs were identified through the distinction between non-CAP epochs that were clearly defined and all other epochs that contained CAP-related activity. The non-CAP epochs consisted of “SLEEP-REM” (2335 epochs) and “SLEEP-S0” (2141 epochs), which made a total of 4476 epochs. The total number of CAP epochs equaled 15,114 based on the remaining epochs after subtracting 4476 non-CAP epochs from 19,590 total epochs.

We calculated the CAP rate by dividing the number of CAP epochs by the total epochs and then multiplying by 100 to express the result as a percentage using these figures. The calculated CAP rate of approximately 77.15% significantly exceeded the clinical threshold of 30%, explicitly indicating pathological sleep patterns within the evaluated samples. Such a high CAP rate is indicative of sleep instability or disrupted sleep architecture, commonly associated with various sleep disorders, including sleep-disordered breathing (SDB), insomnia, and periodic limb movements. Our results thus demonstrate the LSTM–PSO model’s robust capability to objectively identify pathological samples characterized by elevated CAP rates, underscoring its potential as a valuable clinical tool for automated and reliable detection of sleep abnormalities.

The model provided diagnostic support for SDB and insomnia through detailed differentiation and accurate classification of EEG-based cyclic alternating pattern (CAP) phases. The sleep disorders sleep-disordered breathing and insomnia showed specific sleep microstructure disruptions through changes in CAP dynamics. These disorders showed three main features, which included elevated CAP rates, abnormal A1–A3 CAP subtype distribution, and increased EEG arousals. Our experimental setup employed a Long Short-Term Memory network optimized with LSTM–PSO to explicitly classify CAP subtypes. The model achieved robust classification performance, obtaining ROC–AUC scores ≥0.92 for distinguishing CAP subtypes (A1, A2, A3), indicating its strong ability to discriminate subtle EEG differences typical of pathological sleep states.

The elevated CAP rate functions as objective evidence that proves that disrupted sleep microstructure leads to conditions such as SDB and insomnia. The automatic EEG dynamic quantification method eliminated the subjective errors and inter-rater variability that traditional visual analysis methods produce. The diagnostic support provided by our approach delivered clear and objective results at a fast pace, which can help clinicians detect these disorders early. The model’s ability to diagnose sleep disorders needs further validation through clinical annotation of datasets such as the Apnea-ECG Database to establish its practical clinical use. The process of manual CAP annotation takes 3–4 h to complete for a full-night EEG because it involves reviewing multiple channels and segmenting events. The model performed classification and CAP mapping in less than 3 min per record through GPU inference. The time reduction exceeds 90%, which makes high-throughput analysis possible and would help sleep labs that need to operate with reduced staff numbers.

## 7. Conclusions

In our research project, we investigated how advanced ML and DL methods can enhance the analysis and categorization of sleep stages based on CAP EEG signals. The main goal of this study was to utilize LSTM networks that were fine-tuned using PSO, along with the Keras Tuner–Hyperband approach, in order to effectively manage the intricacies linked to sleep stage information. The combination of LSTM with optimization methods like PSO and Hyperband represents a breakthrough in the realm of sleep study research. These techniques have enabled the creation of a model that can accurately differentiate between sleep phases. The model’s effectiveness was thoroughly tested on data obtained from the CAP Sleep Database, which contains a range of recordings from both individuals in good health and those with sleep disorders. Our findings showed that the enhanced LSTM model performed well in terms of accuracy and AUC scores for the categories and proved its effectiveness in categorizing sleep stages. The significant AUC values for SLEEP-REM and SLEEP-SO stages particularly emphasize the precision of the model in identifying these sleep phases. Nevertheless, the model faced difficulties in discerning between intertwined MCAP stages, possibly due to inadequate capture of subtle physiological variations by the current model structure or dataset employed. Moreover, the research validated the significance of model improvement and the necessity for detailed data to enhance classification precision. Subsequent studies could concentrate on broadening the dataset, incorporating multimodal data, and investigating innovative neural network designs to boost the model’s sensitivity and specificity. Future work will involve working with clinical centers to obtain externally annotated CAP datasets or to develop standardized annotations on existing public EEG databases using internationally recognized CAP scoring criteria. We also plan to evaluate the generalizability and transferability of the proposed framework to other sleep staging tasks and more diverse patient populations, including those with specific neurological and sleep-related disorders.

## Figures and Tables

**Figure 1 brainsci-15-00854-f001:**
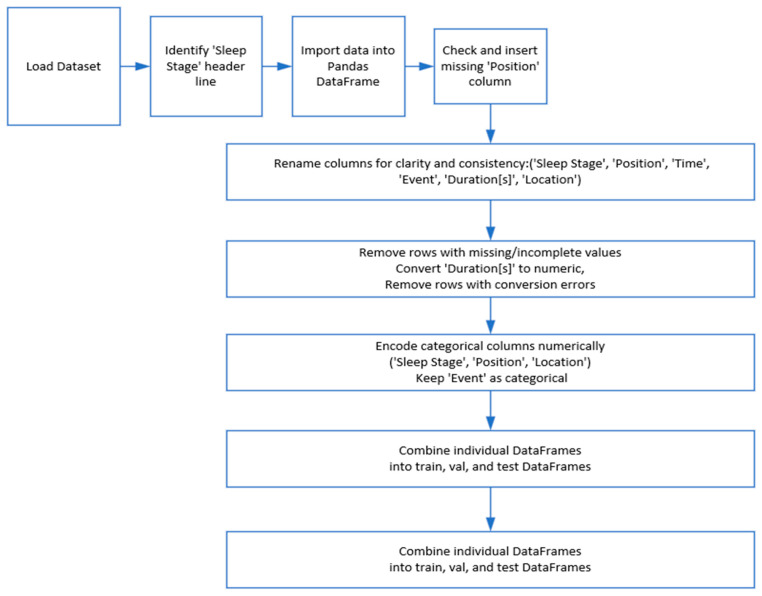
Data preprocessing stages.

**Figure 2 brainsci-15-00854-f002:**
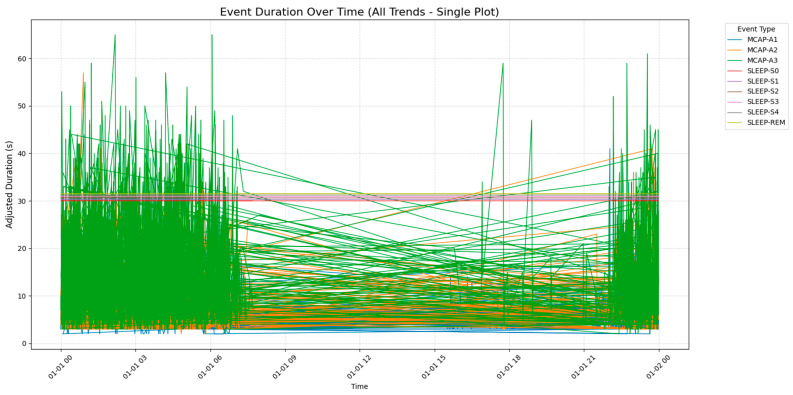
Event duration over time for different sleep stages and CAP phases, illustrating the transitions and durations of sleep stages (SLEEP-S0 to SLEEP-S4, SLEEP-REM) and CAP phases (MCAP-A1 to MCAP-A3) over a 24 h period.

**Figure 3 brainsci-15-00854-f003:**
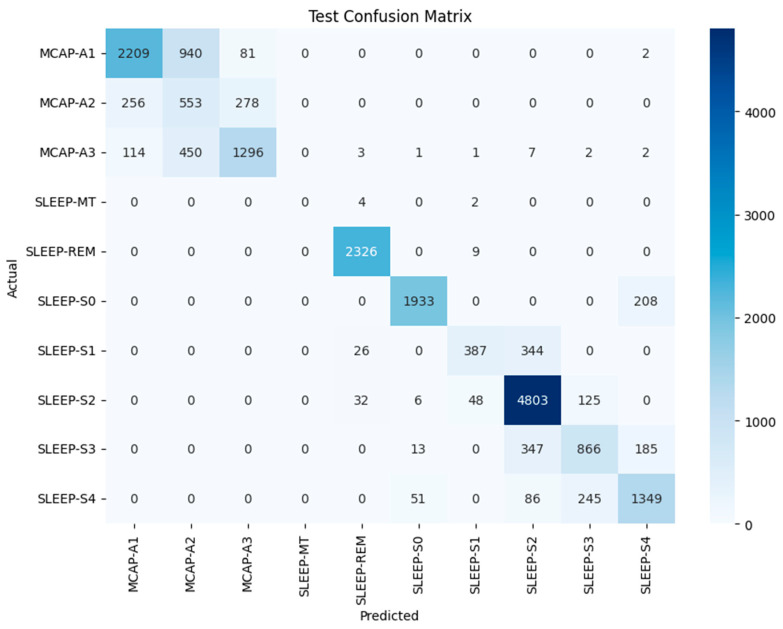
CM of the LSTM model with optimized hyperparameters for test dataset classification across various sleep stages and events.

**Figure 4 brainsci-15-00854-f004:**
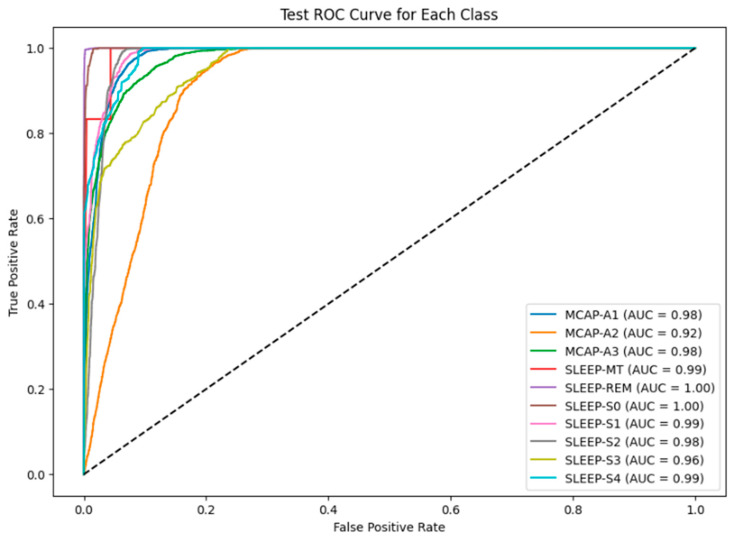
ROC AUC curve for each class in the test dataset.

**Figure 5 brainsci-15-00854-f005:**
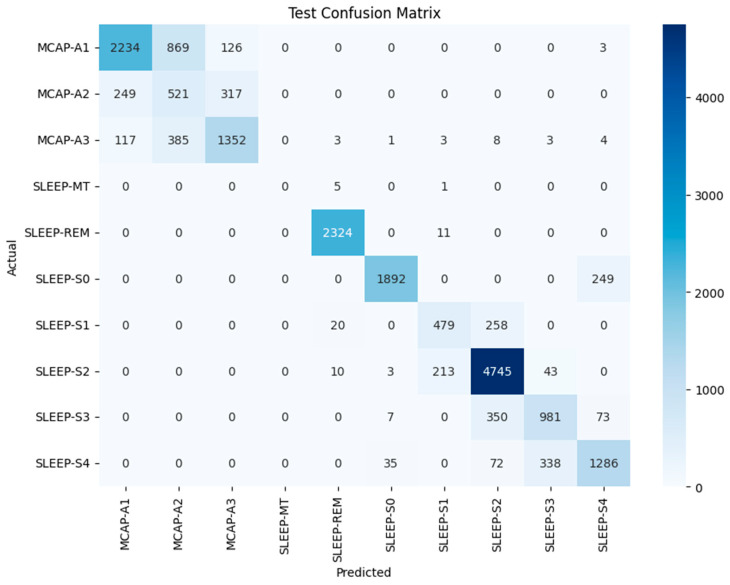
CM of the LSTM model with PSO for test dataset classification across various sleep stages and events.

**Figure 6 brainsci-15-00854-f006:**
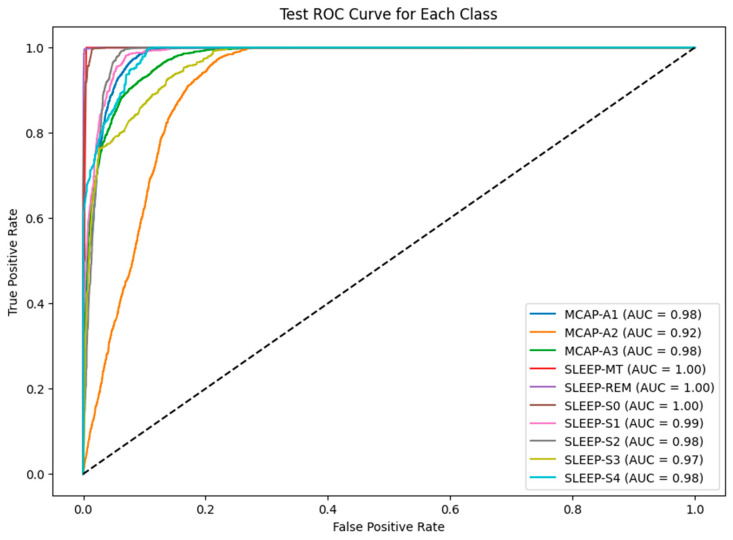
ROC AUC curve for each class in the test dataset for the LSTM-PSO model.

**Figure 7 brainsci-15-00854-f007:**
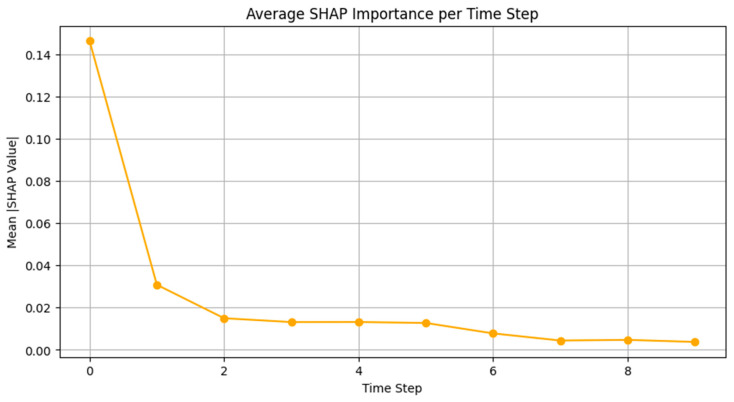
The average SHAP values per time step.

**Figure 8 brainsci-15-00854-f008:**
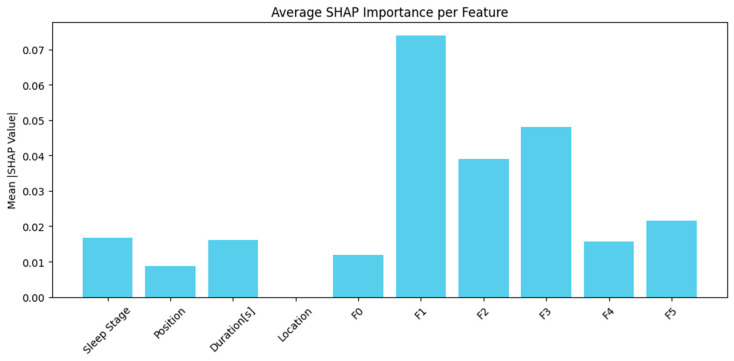
Features combined across time to identify which features most affected predictions.

**Figure 9 brainsci-15-00854-f009:**
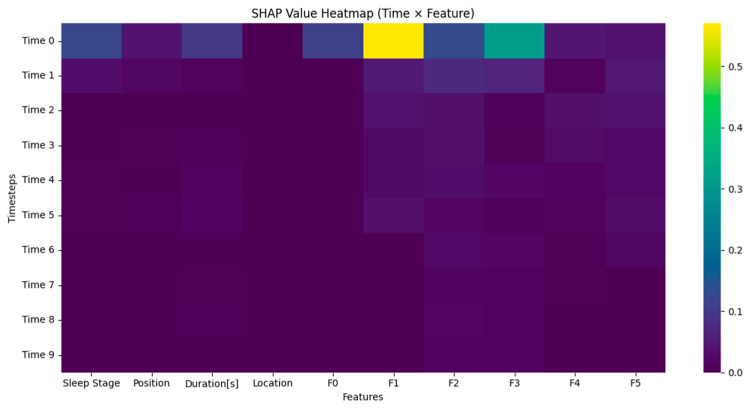
Temporal and feature-level contributions by combining time and feature axes.

**Figure 10 brainsci-15-00854-f010:**
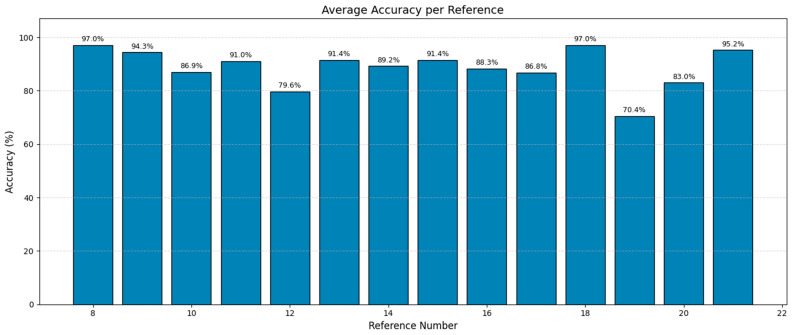
State-of-the-art performance on various datasets.

**Table 1 brainsci-15-00854-t001:** Test classification report for sleep stage detection (LSTM with hyperparameters).

Class	Precision	Recall	F1-Score	Support
SLEEP-REM	0.97	1.00	0.98	2335
SLEEP-S0	0.96	0.90	0.93	2141

**Table 2 brainsci-15-00854-t002:** Optimized hyperparameters and class weights.

Parameter	Method Used	Optimized Range/Value	Purpose
LSTM layers	PSO + Hyperband	1–3	Deep sequence modeling
LSTM units	PSO	64–128	Temporal feature extraction
Dropout rate	PSO	0.2–0.5	Prevents overfitting
Dense units	PSO	64	Time step-level prediction
Batch size	Hyperband	32–128	Training speed vs stability
Learning rate	ReduceLROnPlateau	Starts ~ 0.001	Smooth convergence
Loss function	Fixed	Categorical cross-entropy	Multi-class support
Optimizer	Fixed	Adam	Adaptive learning
Class Weights	Manual (inverse frequency)	Auto-calculated from sample counts	Handle class imbalance (e.g., MCAP-A2 underrep.)

**Table 3 brainsci-15-00854-t003:** CAP = A subtypes (LSTM-Hyperband vs. LSTM-PSO).

CAP Subtype	LSTM-Hyperband Precision (%)	LSTM-Hyperband Recall (%)	LSTM-PSO Precision (%)	LSTM-PSO Recall (%)
MCAP-A1	85	68	97	98
MCAP-A2	28	50	97	97
MCAP-A3	78	69	97	97

**Table 4 brainsci-15-00854-t004:** Ablation study results: PSO versus traditional hyperparameter optimization methods in CAP detection.

Optimization Method	Accuracy (%)	ROC-AUC (REM)	ROC-AUC (MCAP Subtypes)	Precision (MCAP Subtypes)	Recall (MCAP Subtypes)	Computational Efficiency
LSTM+ RF-PCA	82	0.93	0.82	0.68	67	Data loading time 38.44 s PSO + RF-PCA Optimization time 2822.20 s (47.04 min) Training time 243.26 s (4.05 min) Evaluation time 22.5 s Total runtime 3126.40 s (~52.11 min) Samples per second (total) 1,355,852.75 Memory usage peak 2262.90 MB
LSTM + PSO only	97	90	96	96	96	Total Runtime: 11,509.52 s (~191.83 min) Data loading time: 38.44 s PSO optimization time: 11,289.10 s (~188.15 min) Training time: 173.02 s (~2.88 min) Evaluation time: 6.37 s Samples per second (total): 135,699.03 Memory usage peak: 3017.20 MB
LSTM + random search	85	0.95	0.83	0.65	0.66	Moderate: faster optimization (4000 s), better throughput (148,696 samples/s), lower total runtime (71.74 min)
LSTM + grid search	87	0.96	0.85	0.67	0.68	Slower: longer optimization (16,200 s), lower throughput (112,696 samples/s), higher total runtime (275.07 min)

**Table 5 brainsci-15-00854-t005:** CAP feature and CAP subtype.

Feature	Frequency	Description	CAP Subtype Association
F1	Delta (0.5–4 Hz)	Slow activity deep	Subtype A1
F2	Theta (4–8 Hz)	EEG activity intermediate	Subtype A2
F3	Alpha/beta (8–30 Hz)	Lighter sleep states	Subtype A3

**Table 6 brainsci-15-00854-t006:** Summary of AI techniques and their performance results in sleep and neurological studies.

Ref. No.	Methodology	Dataset	Accuracy (%)	ROC-AUC	F1-Score
[8]	WSN + TNN	CAP Sleep, SDRC	97	0.98	0.97
[9]	MML-DMS (VGG16 CNNs)	CAP Sleep (PhysioNet)	94.34	Not reported	0.92
[10]	CNN + Grad-CAM	CHAT	86.9	Not reported	0.82
[11]	EEG-based ML (RBD detection)	iRBD Dataset	91	Not reported	0.963
[12]	LANMAO recorder	Neonatal EEG	79.6	Not reported	0.7693
[13]	SVM (OvA/OvO)	Apnea/Insomnia Dataset	91.44	0.927	0.914
[14]	Ensemble (stacking/voting)	Sleep-EDF	89.2	0.901	0.82
[15]	Bi-LSTM + CNN	Sleep-EDF-20/78	91.44	Not reported	0.8869
[16]	SCORE-AI (CNN)	30,493 EEG records	88.3	0.89	0.81
[17]	CNN-CRF (SleepXAI)	Sleep-EDF	86.8	Not reported	0.86
[18]	SlumberNet (ResNet)	Mouse EEG/EMG	97	Not reported	0.96
[19]	Random Survival Forest	iRBD Dataset	70.4	0.775	0.63
[20]	Random Forest	SEED-VIG	83	Not reported	0.79
[21]	RF + PCA	EEG Dataset	95.2	Not reported	0.82
[22]	CNN + LSTM	HeadbandSleepScorer	76.98	Not available	Not available
[23]	Multi-resolution convolutional neural network (MRCNN)	Sleep-EDF-20	85.6	Not available	80.9
[24]	BiLSTM	PhysioNet Sleep EDF Exp.	83.78	1	82.14
[25]	Fully convolutional neural network	DCSM	86.5	Not available	0.9
[26]	Fully convolutional neural network	Sleep-EDF and SHHS	84.8	Not available	78.8
[27]	CNN	Multiple datasets	86.3	88.8	80.75
[28]	FlexSleepTransformer	Multiple datasets	85		
[29]	Convolutional + recurrent layers	Elderly sleep apnea patients	85	0.77	Not available
[30]	Transfer learning (TL)	Multi-site US PSG database		0.96	
[31]	Vision Transformer (ViT)	SHHS	82.49	Not available	81.94
[32]	Transformer encoders and attention	SHHS	89.7	0.852	0.831
[33]	MultiSEss (Multi-Scale CNN + SE + SSM)	Sleep-EDF-20	83.84	Not available	Not available
[34]	CatBoost, LightGBM, RF	Sleep-EDF-20	86.3	Not available	80.9
[35]	Transformer	SHHS		0.99	
[36]	DL	Sleep-EDF	Not available	Not available	Not available
[37]	Multi-scale Attention Residual Network	Sleep-EDF	84.24	Not available	Not available
[38]	RGMNet	Elderly (60–90 y) Human Sleep Project PSGs	86.61	Not available	Not available
ours	LSTM-PSO-optimized	CAP Sleep Database	97.0	0.98	0.98

## Data Availability

The Apnea-ECG Database is available at https://physionet.org/content/apnea-ecg/1.0.0/ (accessed on 14 November 2024).

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
