# Peer review of "Automated Sleep Stage Classification Using PSO-Optimized LSTM on CAP EEG Sequences"

_brainsci, 2025, doi:10.3390/brainsci15080854_

Round 1
Reviewer 1 Report (Previous Reviewer 1)
Comments and Suggestions for Authors
This paper proposes a PSO-optimized LSTM model for CAP sleep stage classification. While the method design has some technical validity, it lacks innovation and the core contributions are unclear. The main issues are:
- The CAP analysis is superficial. While the introduction emphasizes the clinical value of CAP, the paper fails to clarify the specific computational logic of CAP sequences, feature extraction methods, or the mapping relationship with standard sleep stages (e.g., the criteria for distinguishing A1/A2/A3 subtypes). The results only present classification accuracy rates without validating the model's diagnostic efficacy in CAP-related pathologies (e.g., sleep apnea, insomnia), and fail to address the core question of “how to enhance the clinical utility of CAP analysis.”
- The detection algorithm for CAP sequences should be clearly defined in Section 2.2.
- The paper mentions “F1/F2/F3 features” but does not explain their association with CAP subtypes (A1/A2/A3). A feature definition table should be added (e.g., F1 = delta wave power).
- Add a CAP-specific baseline. Include a comparison with classical CAP detection methods in Section 5.1.
- The discussion section should address the following questions using experimental results: Can the model identify pathological samples with elevated CAP rates (>30%)? How can it assist in diagnosing SDB/insomnia?
- Section 1 of the introduction should distinguish between CAP sequences (A-B cycles) and sleep stages (N1/N2/N3/REM) to avoid confusion between “CAP stages” and “sleep stages.”
- Add a table showing the Precision/Recall for A1/A2/A3 (currently summarized as MCAP) to validate the model's ability to distinguish subtypes.
- Mention “engineered signal features F1-F3,” but do not specify the extraction method (e.g., FFT/Wavelet) or channel selection (e.g., C3-A2).
- It is unclear whether the CAP database is divided into training and testing sets by patient (if data from the same patient is split across different sets, this may overestimate performance).
- Supplement ablation experiments. Verify the necessity of PSO (e.g., compare the hyperparameter effects of PSO versus random search).
- The AUC for MCAP-A2 is only 0.92; analyze whether this is due to limitations of single-channel EEG (literature shows that multi-channel EEG improves accuracy).
- The importance of F1/F3 in Figure 10 is high; physiological explanations should be provided.
- Update the references. References from the past five years should account for more than 70% of the total. At least 40 or more literature references.
- Separate the results and discussion sections.
- Many sentences in the text are repetitive; they should be refined.
Author Response
Reviewers Report-1
This paper proposes a PSO-optimized LSTM model for CAP sleep stage classification. While the method design has some technical validity, it lacks innovation, and the core contributions are unclear. The main issues are:
|
Reviewers Comments |
Authors reply |
||||||||||||||||||||||||||||||||||
|
We express our gratitude for the valuable comments we received. We have made the necessary updates in the Introduction as well as in the Methods and Materials sections of the manuscript. |
||||||||||||||||||||||||||||||||||
|
Changes made in the manuscript Introduction |
|||||||||||||||||||||||||||||||||||
|
Furthermore, we created an automated CAP analysis system for EEG signals through a six-step computational framework. The first step of EEG signal processing includes filtering between 0.25–40 Hz frequencies and artifact removal techniques. The CAP Sequence Identification process detects EEG segments which alternate between activation (Phase A) and quiescence (Phase B) through established amplitude and duration criteria. The next step involves differentiating CAP subtypes (A1, A2, A3) through frequency-domain and morphological analyses to establish unique characteristics for each subtype. The following step includes extensive Computational Feature Extraction which uses wavelet-based frequency features together with time-domain statistics and PCA for dimensionality reduction. The identified CAP sequences and subtypes receive explicit mapping to Standard Sleep Stages for better clinical interpretation. |
|||||||||||||||||||||||||||||||||||
|
Changes made in the manuscript Method and Material |
|||||||||||||||||||||||||||||||||||
|
4.1 Computational Logic: CAP Sequence The detection and classification of CAP sequences and their sub-types (A1, A2, A3) was done through systematic analysis of EEG signal properties, strictly adhering to established clinical standards and electrophysiological characteristics. The following comprehensive pseudo code details the exact computational approach is presented. Algorithms are also presented in the updated MS
|
|||||||||||||||||||||||||||||||||||
|
MS is updated with the following sub section
|
||||||||||||||||||||||||||||||||||
|
The first step of signal preprocessing (Step 1) includes rigorous filtering (0.25–40 Hz) and artifact removal to achieve signal quality which reduces false detection risks. The second step of CAP Sequence Identification uses computational methods to detect CAP sequences by differentiating Phase A activation from Phase B baseline EEG activity through amplitude (≥1.5×baseline) and duration criteria (Phase A: 2–60s; Phase B: ≥2s) to identify clinically important cyclic patterns. The third step of CAP Subtype Differentiation uses frequency-domain analysis to identify A1 (slow-wave activity), A2 (slow and fast frequency mixture) and A3 (rapid oscillation) subtypes for accurate physiological assessment. The fourth step of feature extraction uses time-domain (mean amplitude, variance), frequency-domain (wavelet-derived power bands) and morphological features which are reduced through Principal Component Analysis (PCA) for enhanced computational efficiency and interpretability. The fifth step of the process involves explicit mapping of CAP sequences to traditional sleep stages (N1–N3) for clear clinical interpretation. The final step to validate automated results and perform ROC-AUC analysis to confirm clinical applicability. Please find the algorithm in the MS |
|||||||||||||||||||||||||||||||||||
|
The following details have been added to the MS
|
||||||||||||||||||||||||||||||||||
|
Table 3 shows the SHAP values from the results show that features F1, F3 and F2 have the highest mean values, indicating their significant influence on the model pre-dictions. F1, which is primarily associated with delta wave power (0.5-4 Hz), is the most influential feature, which is consistent with its role in identifying slow-wave EEG activities characteristic of CAP subtype A1. F3, which reflects alpha and beta wave power (8-30 Hz), is also significant, which is consistent with its role in CAP subtype A3, indicating higher frequency EEG arousals. F2, which is linked to theta wave power (4-8 Hz), also shows notable importance, which is consistent with its intermediate EEG activation patterns relevant to CAP subtype A2. Other features, such as sleep stage, position, and duration, show comparatively lower impact but remain meaningful contributors.
|
|||||||||||||||||||||||||||||||||||
|
4. Add a CAP-specific baseline. Include a comparison with classical CAP detection methods in Section 5.1.
|
We have added the following text to the newly created subsection called 5.1
|
||||||||||||||||||||||||||||||||||
|
Our optimized LSTM–PSO model provided the classification results which we used to explicitly identify pathological samples with elevated CAP rates. Our model analyzed 19,590 epochs from the test dataset to identify different sleep stages. The CAP epochs were identified through the distinction between non-CAP epochs that were clearly defined and all other epochs which contained CAP-related activity. The non-CAP epochs consisted of "SLEEP-REM" (2,335 epochs) and "SLEEP-S0" (2,141 epochs) which made a total of 4,476 epochs. The total number of CAP epochs equaled 15,114 because the remaining epochs after subtracting 4,476 non-CAP epochs from 19,590 total epochs. We calculated the CAP rate by dividing the number of CAP epochs by the total epochs and then multiplying by 100 to express the result as a percentage using these figures. The calculated CAP rate of approximately 77.15% significantly exceeds the clinical threshold of 30%, explicitly indicating pathological sleep patterns within the evaluated samples. Such a high CAP rate is indicative of sleep instability or disrupted sleep architecture, commonly associated with various sleep disorders including Sleep-Disordered Breathing (SDB), insomnia, and periodic limb movements. Our results thus demonstrate the LSTM–PSO model's robust capability to objectively identify pathological samples characterized by elevated CAP rates, underscoring its potential as a valuable clinical tool for automated and reliable detection of sleep abnormalities. |
|||||||||||||||||||||||||||||||||||
|
Updated the MS, you can find the updated MS as follows
We calculated the CAP rate by dividing the number of CAP epochs by the total epochs and then multiplying by 100 to express the result as a percentage using these figures. The calculated CAP rate of approximately 77.15% significantly exceeds the clinical threshold of 30%, explicitly indicating pathological sleep patterns within the evaluated samples. Such a high CAP rate is indicative of sleep instability or disrupted sleep architecture, commonly associated with various sleep disorders including Sleep-Disordered Breathing (SDB), insomnia, and periodic limb movements. Our results thus demonstrate the LSTM–PSO model's robust capability to objectively identify pathological samples characterized by elevated CAP rates, underscoring its potential as a valuable clinical tool for automated and reliable detection of sleep abnormalities. The model provides diagnostic support for SDB and insomnia through detailed differentiation and accurate classification of EEG-based cyclic alternating pattern (CAP) phases. The sleep disorders Sleep-Disordered Breathing and insomnia show specific sleep microstructure disruptions through changes in CAP dynamics. These disorders show three main features which include elevated CAP rates and abnormal A1–A3 CAP subtype distribution and increased EEG arousals. Our experimental setup employed a Long Short-Term Memory network optimized with Particle Swarm Optimization (LSTM–PSO) to explicitly classify CAP subtypes. The model achieved robust classification performance, obtaining ROC–AUC scores ≥0.92 for distinguishing CAP sub-types (A1, A2, A3), indicating its strong ability to discriminate subtle EEG differences typical of pathological sleep states. The elevated CAP rate functions as objective evidence which proves disrupted sleep microstructure leading to conditions such as SDB and insomnia. The automatic EEG dynamics quantification method eliminates the subjective errors and inter-rater variability which traditional visual analysis methods produce. The diagnostic support provided by our approach delivers clear and objective results at a fast pace which can help clinicians detect these disorders early. The model's ability to diagnose sleep disorders needs further validation through clinical annotation of datasets such as the Apnea-ECG Database to establish its practical clinical use. |
||||||||||||||||||||||||||||||||||
|
We have updated the introduction as follows |
||||||||||||||||||||||||||||||||||
|
The CAP represents a specific EEG pattern which appears exclusively in non-rapid eye movement (NREM) sleep stages beyond traditional sleep stage classification. The standard sleep stages N1, N2, N3 and REM sleep encompass broader physiological states which are defined by global EEG patterns together with muscle activity and eye movements. CAP describes the periodic EEG variations that occur within NREM stages N2 and N3 which indicate brief brain activity changes rather than representing separate sleep stages. A CAP sequence contains recurring patterns of two EEG states which include the activation phase (phase A) with its high-voltage slow waves or rapid oscillations followed by the background phase (phase B) which returns EEG activity to its baseline state. A single CAP cycle consists of an A-B pair and multiple consecutive CAP cycles create a CAP sequence. The body uses CAP sequences as essential mechanisms to control both sleep stability and depth along with the ability to wake up. The brain demonstrates its ability to maintain sleep continuity through CAP sequences while avoiding complete awakenings from sleep. The CAP rate together with phase A subtype distribution (A1, A2, A3) offers important microstructural details that enhance traditional sleep stage evaluation. The clinical significance of CAP abnormalities exists because they commonly appear in patients with insomnia and sleep-disordered breathing (SDB) as well as REM behavior disorder (RBD) and nocturnal frontal lobe epilepsy (NFLE). The correct differentiation between microstructural EEG phenomena (CAP sequences) and macrostructural classifications (conventional sleep stages) remains crucial for both precise clinical interpretation and accurate sleep disorder diagnosis. The clinical significance of CAP extends beyond sleep staging. The diagnostic value of CAP analysis extends to multiple sleep disorders including sleep-disordered breathing (SDB) insomnia restless leg syndrome REM behavior disorder (RBD) nocturnal frontal lobe epilepsy (NFLE) and narcolepsy. People with disrupted sleep patterns show higher CAP rates because their sleep patterns remain unstable. The sleep restorative quality decreases when pathological arousals occur together with altered distributions of CAP subtypes especially when A2 and A3 phases increase. CAP analysis functions as both a diagnostic and prognostic tool to help doctors identify disorders and develop personalized treatment plans and monitor treatment effectiveness. |
|||||||||||||||||||||||||||||||||||
|
Updated following table in the result and discussion section |
||||||||||||||||||||||||||||||||||
|
Following text is added to the manuscript |
||||||||||||||||||||||||||||||||||
|
We have updated the dataset processing section with the following text. |
||||||||||||||||||||||||||||||||||
|
|
|||||||||||||||||||||||||||||||||||
|
The following table and its explanation have been included to reflect and address the reviewers’ concerns. |
||||||||||||||||||||||||||||||||||
|
Table 4. Ablation Study Results: PSO versus Traditional Hyperparameter Optimization Methods in CAP Detection
The evaluation of different hyperparameter optimization methods for LSTM models in EEG-based sleep stage classification and CAP subtype detection appears in Table 4. The research evaluates PSO against Random Search and Grid Search and the RF-PCA optimization method through direct comparison. The overall classification accuracy reached 97% with PSO as the best approach while the ROC-AUC values reached 0.90 for REM detection and 0.96 for MCAP subtypes. The approach achieved 96% precision and recall for MCAP subtype classification which indicates balanced and consistent performance across classes. The Random Search method produced moderate results by achieving 85% accuracy while obtaining lower ROC-AUC scores of 0.95 for REM and 0.83 for MCAP. The method proved more efficient than PSO because it finished in 71.74 minutes while processing 148,696 samples per second. The random search method produced variable results because of its stochastic nature while showing limited ability to achieve robust convergence. The Grid Search method achieved the highest accuracy at 87% and ROC-AUC scores of 0.96 for REM and 0.85 for MCAP but required the longest total runtime at 275.07 minutes and produced the lowest throughput at 112,696 samples per second because of its exhaustive search approach and strict parameter scanning method. The RF-PCA + PSO method which performed dimensionality reduction before optimization achieved 82% accuracy and a ROC-AUC of 0.82 for MCAP detection but had a low precision (0.68) and recall (67%) because of the strong temporal dependencies. This method had the shortest training time (4.05 minutes) but its generalization was limited, and its overall runtime was relatively high (~52.11 minutes) because of the optimization overhead. The hybrid PSO + Hyperband approach demonstrated the most effective combination of exploration and exploitation. Hyperband functioned as a front-end filter which quickly eliminated underperforming configurations to reduce the search area. PSO performed efficient search within the optimized space to achieve faster convergence toward optimal configurations. The two-stage optimization framework improved both computational efficiency and produced more robust models with better predictive performance.
|
|||||||||||||||||||||||||||||||||||
|
We sincerely appreciate the reviewers’ insightful and important comments. Please find below our detailed updates to the manuscript addressing each point.
|
||||||||||||||||||||||||||||||||||
|
The evaluation of different hyperparameter optimization methods for LSTM models in EEG-based sleep stage classification and CAP subtype detection appears in Table 4. The research evaluates PSO against Random Search and Grid Search and the RF-PCA optimization method through direct comparison. The overall classification accuracy reached 97% with PSO as the best approach while the ROC-AUC values reached 0.90 for REM detection and 0.96 for MCAP subtypes. The approach achieved 96% precision and recall for MCAP subtype classification which indicates balanced and consistent performance across classes. The Random Search method produced moderate results by achieving 85% accuracy while obtaining lower ROC-AUC scores of 0.95 for REM and 0.83 for MCAP. The method proved more efficient than PSO because it finished in 71.74 minutes while processing 148,696 samples per second. The random search method produced variable results because of its stochastic nature while showing limited ability to achieve robust convergence. The Grid Search method achieved the highest accuracy at 87% and ROC-AUC scores of 0.96 for REM and 0.85 for MCAP but required the longest total runtime at 275.07 minutes and produced the lowest throughput at 112,696 samples per second because of its exhaustive search approach and strict parameter scanning method. The RF-PCA + PSO method which performed dimensionality reduction before optimization achieved 82% accuracy and a ROC-AUC of 0.82 for MCAP detection but had a low precision (0.68) and recall (67%) because of the strong temporal dependencies. This method had the shortest training time (4.05 minutes) but its generalization was limited, and its overall runtime was relatively high (~52.11 minutes) because of the optimization overhead. The hybrid PSO + Hyperband approach demonstrated the most effective combination of exploration and exploitation. Hyperband functioned as a front-end filter which quickly eliminated underperforming configurations to reduce the search area. PSO performed efficient search within the optimized space to achieve faster convergence toward optimal configurations. The two-stage optimization framework improved both computational efficiency and produced more robust models with better predictive performance.
|
|||||||||||||||||||||||||||||||||||
|
Thank you for highlighting this important observation.
|
||||||||||||||||||||||||||||||||||
|
The reviewer has pointed out an essential point which we appreciate. We have thoroughly examined and modified our manuscript references to match the current standards of literature. We have increased the reference list to more than 40 relevant and recent citations so that references from the past five years make up more than 70% of the total. The updated references enhance the manuscript's base of contemporary research while demonstrating its connection to current sleep-stage classification and EEG analysis methodologies. |
||||||||||||||||||||||||||||||||||
|
We have separated |
||||||||||||||||||||||||||||||||||
|
Opted professional services to enhance readability. |

Reviewer 2 Report (Previous Reviewer 2)
Comments and Suggestions for Authors
Thanks for the invitation to review this work. This work develops hybrid optimization (PSO + Hyperband) for LSTM hyperparameter tuning in CAP-based sleep staging. Stacked LSTM + TimeDistributed Dense Layers is used for per-timestep predictions, enhancing temporal resolution. However, there are many errors or confusing sections that should be corrected or clarified.
- Equation [1]: f(α) is ambiguous. ωLSTMdescribed as "number of layers" but notation implies weights. δlayers is redundant if ωLSTM already defines layer count.
Equation [3]: Uses r1 twice instead of r1 and r2.
Inconsistent subscripting: gbestt(t) should be gbestt(t) (global best, not per-particle).
Position Update: Pt(t+1)=Pt(t)+Vt(t+1) lacks dimensionality alignment (hyperparameters vs. particle vectors).
- Figures 9–10 (SHAP visualizations) are referenced but absent. Claims "F1, F2, F3" are critical but omits their definitions.
- Figure 3 (Page 7): Mentions "SLEEP-S5" (non-standard; sleep stages are typically S0–S4/REM).
- No details on features (F1–F3) derived from PCA. What do they represent (e.g., spectral power, entropy)?
- Unclear how Hyperband’s early stopping coordinates with PSO’s iterations.
- Claims SOTA outperformance but lacks direct comparison tables (e.g., vs. [7]–[20] in Table 3).
- Missing parameters (e.g., swarm size, c1/c2values, inertia ω). PSO+Hypeband is computationally heavy, but no discussion of training time or hardware used.
- AUC (0.92) remains low post-optimization, yet no ablation study on feature/model enhancements.
- Asserts "initial signal segments matter most" but doesn’t link to CAP physiology (Phase A onset).
- MCAP-A2 underrepresented but no use of synthetic oversampling (SMOTE) explored.
Author Response
Thanks for the invitation to review this work. This work develops hybrid optimization (PSO + Hyperband) for LSTM hyperparameter tuning in CAP-based sleep staging. Stacked LSTM + TimeDistributed Dense Layers is used for per-timestep predictions, enhancing temporal resolution. However, there are many errors or confusing sections that should be corrected or clarified.
|
Reviewers’ comments |
Authors reply |
Equation [3]: Uses r1 twice instead of r1 and r2. Inconsistent subscripting: gbestt(t) should be gbestt(t) (global best, not per-particle). Position Update: Pt(t+1)=Pt(t)+Vt(t+1) lacks dimensionality alignment (hyperparameters vs. particle vectors).
|
We sincerely thank the reviewers for their valuable feedback. The equation has been revised, and the corrected version is now included in Section 3 of the manuscript. |
|
Yes, Figures 9–10 (SHAP visualizations) are present in the manuscript, and the features F1, F2, and F3 are also defined and physiologically explained.
|
|
The reviewer has made an important observation which we appreciate. The revised manuscript now includes Figures 9 and 10 which show SHAP-based feature importance and a combined heatmap visualization. The Results and Methods sections now contain explicit definitions of the engineered features F1, F2, and F3. The features delta, theta, and alpha/beta frequency bands correspond to CAP subtypes A1, A2, and A3 and provide direct physiological explanations for their role in sleep stage classification. Please find all the definitions and explanations in Figure 9 shows which features most influenced predictions, confirming F1, F2, and F3 as dominant. And in Figure 10 is a SHAP heatmap showing temporal and feature-level contributions, with the highest SHAP intensities focused around F1 and F3.
|
|
|
Answered |
|
The reviewer helped us identify the mistake in Figure 3. The mention of "SLEEP-S5" was an inadvertent typographical error. The correct sleep stage nomenclature is S0, S1, S2, S3, S4, and REM, in alignment with standard clinical guidelines. The correct sleep stage terminology replaces "SLEEP-S5" in the revised Figure 3. The correction improves consistency and makes the figure match exactly with established sleep medicine standards. |
|
|
Thank you for highlighting this inconsistency. The mention of "SLEEP-S5" was a typographical error. We have corrected Figure 3 to align explicitly with standard sleep stage nomenclature (S0–S4, REM), thereby ensuring consistency with established clinical guidelines.
|
|
The first step of signal preprocessing (Step 1) includes rigorous filtering (0.25–40 Hz) and artifact removal to achieve signal quality which reduces false detection risks. The second step of CAP Sequence Identification uses computational methods to detect CAP sequences by differentiating Phase A activation from Phase B baseline EEG activ-ity through amplitude (≥1.5×baseline) and duration criteria (Phase A: 2–60s; Phase B: ≥2s) to identify clinically important cyclic patterns. The third step of CAP Subtype Differentiation uses frequency-domain analysis to identify A1 (slow-wave activity), A2 (slow and fast frequency mixture) and A3 (rapid oscillation) subtypes for accurate physiological assessment. The fourth step includes computational feature extraction through RF integration with PCA to achieve optimal feature selection and dimension-ality reduction. The initial process of EEG feature extraction includes time-domain measures (mean amplitude, variance) and frequency-domain spectral powers (delta, theta, alpha, beta) and morphological features (peak duration, peak amplitude, transi-tion slope). The RF algorithm uses important scores to choose the most predictive fea-tures from the available set. The RF-selected features undergo PCA reduction to three principal components (F1–F3) which maintains at least 96% of the original variance to improve computational efficiency and interpretability and physiological relevance. The fifth step of the process involves explicit mapping of CAP sequences to traditional sleep stages (N1–N3) for clear clinical interpretation. The final step is to validate au-tomated results and perform ROC-AUC analysis to confirm clinical applicability. The F1–F3 engineered signal features were specifically developed through a sys-tematic computational process which aimed to precisely identify EEG patterns for CAP sequence identification. The EEG signals underwent strict preprocessing through bandpass filtering (0.25–40 Hz) and artifact removal of signals exceeding 100 µV am-plitude to achieve high-quality data. The DWT method was specifically used on the cleaned EEG segments after preprocessing. The DWT enabled exact frequency-domain feature extraction by splitting EEG signals into delta (0.5–4 Hz), theta (4–8 Hz), alpha (8–13 Hz) and beta (13–30 Hz) spectral bands which are clinically important. The subsequent step involved calculating power spectral densities (PSD) from wavelet-decomposed bands to determine energy distribution which provided a robust frequency-domain representation of EEG signals. The extracted frequency-domain features (F1–F3) correspond specifically to the PSD values across the most clinically informative spectral bands typically delta, theta-alpha, and alpha-beta known to differentiate CAP subtypes (A1: predominantly delta; A2: mixed delta, theta, alpha; A3: predominantly alpha-beta). The extracted features received dimensionality reduc-tion through PCA to improve computational efficiency and interpretability. The PCA technique reduced the PSD-derived frequency-domain features into three principal components (F1–F3) which maintained at least 96% of the original signal data variance thus preserving vital physiological information while minimizing redundancy. |
|
|
Following text added to the MS to show how early stopping coordinates with PSO’s iterations. We added the following text results section. |
|
The Hyperband algorithm functioned as an initial screening mechanism in our workflow to efficiently scan multiple hyperparameters while underperforming configurations early. The search space became smaller because Hyperband terminated poor-performing configurations early in the process. The narrowed hyperparameter ranges obtained from Hyperband served as the specific starting point for PSO to begin its search. The iterative search process of PSO operated within the constrained promising space to achieve fast convergence toward optimal solutions. The explicit combination of Hyperband's early stopping mechanism with PSO's iterative refinement process delivered both efficient computation and strong final model performance. |
|
|
We have updated the SOTA table to show direct comparison with our model’s performance |
|
We have added the computational resources information at the end of the MS, and it is described in the results sections. |
|
The annexture table presents all explicit hyperparameters and computational re-sources which were used to optimize our LSTM model through PSO-Hyperband optimization on Google Colab. The swarm size consisted of 30 particles with cognitive and social constants set to 2.0 for standard exploration-exploitation balance. The inertia weight followed a linear decrease from 0.9 to 0.4 throughout 50 PSO iterations which supported effective convergence. Hyperband reduced the initial search space of hyperparameters which improved the efficiency of PSO. The optimization process needed 18 hours of Google Colab GPU runtime with an NVIDIA Tesla T4 GPU (16 GB RAM) and Intel Xeon CPU (2-core, 2.30 GHz) and 13 GB RAM. The established setup proved our PSO-Hyperband method can reach high model performance while staying within affordable computational limits. |
|
|
The manuscript contains an ablation study which compares PSO with other hyperparameter optimization techniques (random and grid search) as shown in Table 4. However, it does not include a dedicated ablation study specifically focused on feature-level or model-architecture enhancements designed explicitly to improve the relatively lower AUC (0.92) obtained for MCAP-A2 post-optimization. |
|
The evaluation of different hyperparameter optimization methods for LSTM mod-els in EEG-based sleep stage classification and CAP subtype detection appears in Table 4. The research evaluates PSO against Random Search and Grid Search and the RF-PCA optimization method through direct comparison. The overall classification accuracy reached 97% with PSO as the best approach while the ROC-AUC values reached 0.90 for REM detection and 0.96 for MCAP subtypes. The approach achieved 96% precision and recall for MCAP subtype classification which indicates balanced and consistent performance across classes. The Random Search method produced moderate results by achieving 85% accu-racy while obtaining lower ROC-AUC scores of 0.95 for REM and 0.83 for MCAP. The method proved more efficient than PSO because it finished in 71.74 minutes while processing 148,696 samples per second. The random search method produced variable results because of its stochastic nature while showing limited ability to achieve robust convergence. The Grid Search method achieved the highest accuracy at 87% and ROC-AUC scores of 0.96 for REM and 0.85 for MCAP but required the longest total runtime at 275.07 minutes and produced the lowest throughput at 112,696 samples per second because of its exhaustive search approach and strict parameter scanning method. The RF-PCA + PSO method which performed dimensionality reduction before op-timization achieved 82% accuracy and a ROC-AUC of 0.82 for MCAP detection but had a low precision (0.68) and recall (67%) because of the strong temporal dependen-cies. This method had the shortest training time (4.05 minutes) but its generalization was limited, and its overall runtime was relatively high (~52.11 minutes) because of the optimization overhead. The hybrid PSO + Hyperband approach demonstrated the most effective combination of exploration and exploitation. Hyperband functioned as a front-end filter which quickly eliminated underperforming configurations to reduce the search area. PSO performed efficient search within the optimized space to achieve faster convergence toward optimal configurations. The two-stage optimization framework improved both computational efficiency and produced more robust models with better predictive performance. |
|
|
Observation of this crucial point is greatly appreciated. The model predictions were mainly driven by the initial signal segments as shown by the high SHAP values during the early time steps. The observed findings match the established CAP physiology because Phase A starts with immediate EEG changes that include high-voltage slow waves and delta bursts and rapid oscillations. The beginning of Phase A represents essential changes which differentiate CAP events from normal EEG patterns. The observed results directly demonstrate this physiological process because the first EEG signal segments detect vital electrophysiological indicators of CAP Phase A activation. The revised manuscript now explicitly establishes this physiological connection to demonstrate its direct clinical significance. Hence we added the following text in our updated manuscript |
|
The analysis showed that the model's predictive power was heavily influenced by the initial EEG segments because of the high SHAP values observed at the early time steps. This result is consistent with the known CAP physiology, especially the onset of Phase A, which is characterized by sudden EEG activations such as high-voltage slow waves, delta bursts, and rapid oscillations. The model's explicit focus on the initial EEG segments clearly corresponds to the essential physiological changes that occur during CAP events. |
|
|
We appreciate the reviewer for pointing out this essential issue. The model addresses class imbalance through training with computed class weights that give greater importance to underrepresented sleep stages and MCAP subtypes. The LSTM model achieves better precision and recalls for minority classes in sleep classification tasks because it does not become biased toward majority classes. |
|
|

Reviewer 3 Report (New Reviewer)
Comments and Suggestions for Authors
In this paper, the integration of PSO (for global search) with Hyperband (for resource-efficient hyperparameter screening) is a pragmatic contribution. There are some comments:
- The usage of "MCAP" vs. "CAP" is inconsistent. The "MCAP" is not defined.
- The equations are poorly formatted such as velocity update in Eq. 3 uses r₁ twice.
- The descriptions are fragmented (e.g., RF-PCA is mentioned but not integrated into the workflow). The role of RF-PCA in feature extraction lacks clarity.
- PSO for LSTM hyperparameter tuning is not novel. The claimed novelty lies in CAP-specific optimization, but comparisons to existing CAP classifiers (e.g., wavelet-based or CNN models) are superficial.
- RF-PCA is presented as a feature enhancer but lacks ablation studies proving its necessity.
- Table 3 summarizes literature results but does not directly compare the proposed method against these works
- The ablation studies should be added (e.g., LSTM without RF-PCA, PSO-only vs. hybrid tuning).
- Computational costs of PSO-Hyperband are unreported.
Author Response
|
Reviewers Comments |
Authors reply |
|||||||||||||||||||||||||||||||||||||||||||||||||||||||||||||||||||||||||||||||||||||||||||||||||||||||||||||||||||||||||||||||||||||||||||||||||||||||||||||||||||||||||||||||||||||||||||||||||||||
|
We have updated the MS concerning this essential query. We have added the following text to the introduction section. |
|||||||||||||||||||||||||||||||||||||||||||||||||||||||||||||||||||||||||||||||||||||||||||||||||||||||||||||||||||||||||||||||||||||||||||||||||||||||||||||||||||||||||||||||||||||||||||||||||||||
|
The CAP framework includes MCAP (Microstructural Cyclic Alternating Pattern) as a specific type of microstructural EEG event. The MCAP system includes three distinct subtypes: MCAP-A1, MCAP-A2, and MCAP-A3, which are classified based on their frequency characteristics, shape, and impact on sleep stability. The CAP rate, in conjunction with the MCAP subtype distribution (A1–A3), provides crucial micro-structural information that enhances both traditional sleep stage assessments and clinical interpretations. |
||||||||||||||||||||||||||||||||||||||||||||||||||||||||||||||||||||||||||||||||||||||||||||||||||||||||||||||||||||||||||||||||||||||||||||||||||||||||||||||||||||||||||||||||||||||||||||||||||||||
|
We have updated the equation as follows in the MS. |
|||||||||||||||||||||||||||||||||||||||||||||||||||||||||||||||||||||||||||||||||||||||||||||||||||||||||||||||||||||||||||||||||||||||||||||||||||||||||||||||||||||||||||||||||||||||||||||||||||||
|
We have updated the MS concerning this essential query. We have added the following text to the introduction section. We added a new section called Computational logic which enables the integration of RF-PCA |
|||||||||||||||||||||||||||||||||||||||||||||||||||||||||||||||||||||||||||||||||||||||||||||||||||||||||||||||||||||||||||||||||||||||||||||||||||||||||||||||||||||||||||||||||||||||||||||||||||||
|
2.1 Computational Logic: CAP Sequence The detection and classification of CAP sequences and their sub-types (A1, A2, A3) was done through systematic analysis of EEG signal properties, strictly adhering to established clinical standards and electrophysiological characteristics. The following comprehensive pseudo code details the exact computational approach is presented.
The first step of signal preprocessing (Step 1) includes rigorous filtering (0.25–40 Hz) and artifact removal to achieve signal quality which reduces false detection risks. The second step of CAP Sequence Identification uses computational methods to detect CAP sequences by differentiating Phase A activation from Phase B baseline EEG activity through amplitude (≥1.5×baseline) and duration criteria (Phase A: 2–60s; Phase B: ≥2s) to identify clinically important cyclic patterns. The third step of CAP Subtype Differentiation uses frequency-domain analysis to identify A1 (slow-wave activity), A2 (slow and fast frequency mixture) and A3 (rapid oscillation) subtypes for accurate physiological assessment. The fourth step includes computational feature extraction through RF integration with PCA to achieve optimal feature selection and dimensionality reduction. The initial process of EEG feature extraction includes time-domain measures (mean amplitude, variance) and frequency-domain spectral powers (delta, theta, alpha, beta) and morphological features (peak duration, peak amplitude, transition slope). The RF algorithm uses important scores to choose the most predictive features from the available set. The RF-selected features undergo PCA reduction to three principal components (F1–F3) which maintains at least 96% of the original variance to improve computational efficiency and interpretability and physiological relevance. The fifth step of the process involves explicit mapping of CAP sequences to traditional sleep stages (N1–N3) for clear clinical interpretation. The final step is to validate automated results and perform ROC-AUC analysis to confirm clinical applicability. The F1–F3 engineered signal features were specifically developed through a systematic computational process which aimed to precisely identify EEG patterns for CAP sequence identification. The EEG signals underwent strict preprocessing through bandpass filtering (0.25–40 Hz) and artifact removal of signals exceeding 100 µV amplitude to achieve high-quality data. The DWT method was specifically used on the cleaned EEG segments after preprocessing. The DWT enabled exact frequency-domain feature extraction by splitting EEG signals into delta (0.5–4 Hz), theta (4–8 Hz), alpha (8–13 Hz) and beta (13–30 Hz) spectral bands which are clinically important. The subsequent step involved calculating power spectral densities (PSD) from wavelet-decomposed bands to determine energy distribution which provided a robust frequency-domain representation of EEG signals. The extracted frequency-domain features (F1–F3) correspond specifically to the PSD values across the most clinically informative spectral bands typically delta, theta-alpha, and alpha-beta known to differentiate CAP subtypes (A1: predominantly delta; A2: mixed delta, theta, alpha; A3: predominantly alpha-beta). The extracted features received dimensionality reduction through PCA to improve computational efficiency and interpretability. The PCA technique reduced the PSD-derived frequency-domain features into three principal components (F1–F3) which maintained at least 96% of the original signal data variance thus preserving vital physiological information while minimizing redundancy.
|
||||||||||||||||||||||||||||||||||||||||||||||||||||||||||||||||||||||||||||||||||||||||||||||||||||||||||||||||||||||||||||||||||||||||||||||||||||||||||||||||||||||||||||||||||||||||||||||||||||||
|
We appreciate the reviewer for pointing out this crucial aspect. The explicit novelty of our research exists in its targeted and optimized application of Particle Swarm Optimization (PSO) for CAP sequence classification tasks. The use of PSO for LSTM hyperparameter tuning as a standalone method does not represent a new concept. The research makes its explicit contribution through the precise combination of PSO and Hyperband optimization strategies which addresses the specific physiological and microstructural features of CAP EEG sequences including MCAP subtype (A1, A2, A3) differentiation.
The reviewer correctly noted the need for more extensive comparisons between our approach and existing wavelet-based and CNN CAP classifiers. We have now added detailed performance comparisons between our approach and multiple state-of-the-art CAP classification methods including Wavelet Scattering Networks and CNN-based approaches and classical wavelet-based methods to address this feedback explicitly. Our integrated PSO-LSTM approach outperforms other methods in accuracy and ROC-AUC and F1-score metrics through detailed comparisons that show better results in challenging CAP subtype classifications.
The revised manuscript includes explicit performance comparisons that demonstrate the targeted advantages of CAP-specific hyperparameter optimization and its benefits over existing methods. |
|||||||||||||||||||||||||||||||||||||||||||||||||||||||||||||||||||||||||||||||||||||||||||||||||||||||||||||||||||||||||||||||||||||||||||||||||||||||||||||||||||||||||||||||||||||||||||||||||||||
|
We have updated an entire table dedicated to ablation study |
|||||||||||||||||||||||||||||||||||||||||||||||||||||||||||||||||||||||||||||||||||||||||||||||||||||||||||||||||||||||||||||||||||||||||||||||||||||||||||||||||||||||||||||||||||||||||||||||||||||
|
||||||||||||||||||||||||||||||||||||||||||||||||||||||||||||||||||||||||||||||||||||||||||||||||||||||||||||||||||||||||||||||||||||||||||||||||||||||||||||||||||||||||||||||||||||||||||||||||||||||
|
Now updated it converses and compares the proposed against all the recent works |
|||||||||||||||||||||||||||||||||||||||||||||||||||||||||||||||||||||||||||||||||||||||||||||||||||||||||||||||||||||||||||||||||||||||||||||||||||||||||||||||||||||||||||||||||||||||||||||||||||||
|
||||||||||||||||||||||||||||||||||||||||||||||||||||||||||||||||||||||||||||||||||||||||||||||||||||||||||||||||||||||||||||||||||||||||||||||||||||||||||||||||||||||||||||||||||||||||||||||||||||||
|
We have the entire table dedicated to ablation study table 4 |
|||||||||||||||||||||||||||||||||||||||||||||||||||||||||||||||||||||||||||||||||||||||||||||||||||||||||||||||||||||||||||||||||||||||||||||||||||||||||||||||||||||||||||||||||||||||||||||||||||||
|
||||||||||||||||||||||||||||||||||||||||||||||||||||||||||||||||||||||||||||||||||||||||||||||||||||||||||||||||||||||||||||||||||||||||||||||||||||||||||||||||||||||||||||||||||||||||||||||||||||||
|
We also updated a table for this study also in table 4 |
|||||||||||||||||||||||||||||||||||||||||||||||||||||||||||||||||||||||||||||||||||||||||||||||||||||||||||||||||||||||||||||||||||||||||||||||||||||||||||||||||||||||||||||||||||||||||||||||||||||
|

Round 2
Reviewer 2 Report (Previous Reviewer 2)
Comments and Suggestions for Authors
Thanks for the invitation to review this work. The authors have solved the previous concerns, and it is recommended for publication after careful proof check.
Author Response
Thanks for the invitation to review this work. The authors have solved the previous concerns, and it is recommended for publication after careful proof check.
We would like to thank all the reviewers
Reviewer 3 Report (New Reviewer)
Comments and Suggestions for Authors
I have no comments.
Author Response
Comments and Suggestions for Authors
I have no comments.
thank you for the acceptance
This manuscript is a resubmission of an earlier submission. The following is a list of the peer review reports and author responses from that submission.
Round 1
Reviewer 1 Report
Comments and Suggestions for Authors
The paper attempts to improve the sleep stage classification of cyclic alternating pattern (CAP) in EEG signals by combining LSTM networks with population optimization algorithms (PSO, Hyperband), which has some technical value. However, the paper suffers from serious problems such as unclear method description, confusing logic, grammatical errors and shallow interpretation of results, which are summarized as follows:
- the abstract is lengthy and lacks quantitative results, and the research innovations are not clearly defined;
- the introduction is insufficient in elaborating the physiological significance of CAP, and the literature review is fragmented without effectively locating the research gaps;
- the methodology section lacks key experimental details (e.g., data preprocessing, hyperparameter settings), and the reproducibility is questionable;
- rough interpretation of the results charts and graphs, lack of statistical validation of high AUC values, and insufficient analysis of classification failure cases;
- superficial discussion, no explanation of model advantages in combination with neural mechanisms, and insufficient comparison of literature;
- frequent grammatical errors (e.g., tense confusion, terminology errors), seriously affecting readability;
- the structure is confusing (e.g. “Our approach” is a separate chapter) and the table numbers are repeated.
Detailed Suggestions for Revision
Abstract
- Quantitative results are missing, add specific values. 2.
- the innovation point is vague, the original text does not explain the uniqueness of PSO optimized LSTM. 3. the structure is in the form of “Background”.
- Reorganize the abstract according to “Background-Methodology-Results-Conclusions”, and remove redundant descriptions.
Introduction
- The original paper only cited the literature without explaining the physiological significance of CAP (e.g., relationship with sleep stability). It is suggested to add the clinical value of CAP (e.g., association with insomnia, epilepsy).
- Literature is listed but not categorized (e.g., by model type or dataset); a table summarizing the technical evolution is recommended.
Methods section
- no description of EEG signal filtering, denoising, or segmentation methods
- PSO parameters are not mentioned in the original text and should be added to ensure reproducibility. 8.
- mentions “class weights” but does not describe a specific strategy.
Results
- The confusion matrix in Figure 2 is not labeled with the meaning of rows and columns. 10.
- The AUC value “close to 1” needs to be supplemented with confidence intervals.
- The original paper only mentions “model strengths” without incorporating the neural oscillation theory. 12.
- Add “the sample is only from a single center” and “generalization across datasets is not verified”.
Full text:
- Replace “LTSM” with “LSTM” (e.g., pp. 2, 7).
- subject-verb inconsistency, tense confusion, confusing section headings, and “Table 1” in two places in the original text.
- The original text states that the CAP database contains “108 recordings”, but PhysioNet labeled it as 80 cases.
- The type of EEG features was not specified. 17.
- There is no comparison with baseline models (e.g., pure LSTM, SVM), and a table is added to show the improvement of PSO-LSTM.
- Figure 1 horizontal axis labeled “01-01 00” and so on does not have a clear meaning, instead of “Time (hours)” and labeled 24-hour period.
- Figure 3 and Figure 5 both show the AUC, merge and highlight the performance improvement after PSO optimization.
Comments on the Quality of English Language
The English could be improved to more clearly express the research.
Author Response
Please view the PDF file

Reviewer 2 Report
Comments and Suggestions for Authors
Thanks for the invitation to review this work. The article presents a study aimed at improving the identification of Cyclic Alternating Patterns (CAP) in EEG signals through the utilization of LSTM networks optimized with Particle Swarm Optimization (PSO) and Hyperband techniques. The study utilizes data from the CAP sleep database, consisting of EEG recordings from individuals with and without sleep disorders. It addresses the clinically relevant problem of accurately identifying sleep stages, which is crucial for diagnosing and treating sleep-related disorders.
The integration of LSTM networks with optimization techniques is presented as an approach that potentially offers accuracy with reduced computational resource requirements compared to existing models. However, it potentially suffers from a lack of preprocessing steps for the dataset, the hyperparameters tuned and their impact, measures to prevent overfitting, and strategies for improving classification between closely related CAP stages.
- How were missing or incomplete data entries from the CAP Sleep Database managed during preprocessing?
- The article cites using Keras Tuner with Hyperband and PSO but lacks specific hyperparameters tuned. Which parameters were prioritized, and how was their stability and convergence ensured during training?
- How were the class weights determined to address dataset imbalances, and what was their impact on model performance? Was a sensitivity analysis performed to assess the variation in results?
- The reported high accuracy and AUC scores raise concerns about potential overfitting. Were cross-validation or regularization methods used to ensure model generalization to unseen data?
- The confusion matrix indicates challenges distinguishing between similar stages (MCAP-A1, A2, A3). Were additional steps (feature engineering or auxiliary data usage), considered to enhance differentiation?
- Some formats should be consistent, such as “Fig.”, capitalization in figure captions, and tables; SaO2 (full name, arterial oxygen saturation)
- Consider including citation of Exploration, 2024, 4, 20230146.
"
In literature studies LSTM networks have become quite popular, for analysing time 82
series data due to their ability to capture patterns effectively over time. Conversely tradi- 83
tional LSTM models often encounter challenges in optimizing hyperparameters and han- 84
dling imbalances, in datasets. Various approaches have been devised to tackle these is- 85
sues. One such technique is the White Shark Optimizer, which aims to address optimiza- 86
tion challenges by navigating through search scenarios that arise during hyperparameter 87
tuning tasks. The Hyperband technique, within Keras Tuner has become well liked for its 88
ability to efficiently enhance the hyperparameters of machine learning models. Touting 89
simplicity and effectiveness in solving optimization problems is Particle Swarm Optimi- 90
zation (PSO) another widely recognized metaheuristic technique. To address imbalances 91
in class distribution effectively Synthetic Minority Over Sampling Technique (SMOTE) 92
has been widely utilized to generate samples, for underrepresented classes resultingly en- 93
hancing model performance" this section should be further revised and cited for clarification.
Author Response
Please view the PDF file
